# Trisomy 21 activates the kynurenine pathway via increased dosage of interferon receptors

Rani K. Powers[1,2,3], Rachel Culp-Hill[4], Michael P. Ludwig[1,3], Keith P. Smith[1], Katherine A. Waugh[1], Ross Minter[1], Kathryn D. Tuttle [1], Hannah C. Lewis[1], Angela L. Rachubinski[1,5], Ross E. Granrath [1], María Carmona-Iragui[6,7], Rebecca B. Wilkerson[4], Darcy E. Kahn[1], Molishree Joshi[8], Alberto Lleó[6], Rafael Blesa[6], Juan Fortea[6,7], Angelo D'Alessandro[1,4], James C. Costello[2,3], Kelly D. Sullivan [1,3,5,8]* & Joaquin M. Espinosa[1,3,8,9]*

Trisomy 21 (T21) causes Down syndrome (DS), affecting immune and neurological function by ill-defined mechanisms. Here we report a large metabolomics study of plasma and cerebrospinal fluid, showing in independent cohorts that people with DS produce elevated levels of kynurenine and quinolinic acid, two tryptophan catabolites with potent immunosuppressive and neurotoxic properties, respectively. Immune cells of people with DS overexpress *IDO1*, the rate-limiting enzyme in the kynurenine pathway (KP) and a known interferon (IFN)-stimulated gene. Furthermore, the levels of IFN-inducible cytokines positively correlate with KP dysregulation. Using metabolic tracing assays, we show that overexpression of *IFN* receptors encoded on chromosome 21 contribute to enhanced IFN stimulation, thereby causing *IDO1* overexpression and kynurenine overproduction in cells with T21. Finally, a mouse model of DS carrying triplication of IFN receptors exhibits KP dysregulation. Together, our results reveal a mechanism by which T21 could drive immunosuppression and neurotoxicity in DS.

[1] Linda Crnic Institute for Down Syndrome, University of Colorado Anschutz Medical Campus, Aurora, Colorado, USA. [2] Computational Bioscience Program, University of Colorado Anschutz Medical Campus, Aurora, Colorado, USA. [3] Department of Pharmacology, University of Colorado Anschutz Medical Campus, Aurora, Colorado, USA. [4] Department of Biochemistry and Molecular Genetics, University of Colorado Anschutz Medical Campus, Aurora, Colorado, USA. [5] Department of Pediatrics, University of Colorado Anschutz Medical Campus, Aurora, Colorado, USA. [6] Department of Neurology, Hospital de la Santa Creu i Sant Pau, Biomedical Research Institute Sant Pau, Universitat Autonoma de Barcelona, CIBERNED, Barcelona, Spain. [7] Barcelona Down Medical Center, Catalan Down Syndrome Foundation, Barcelona, Spain. [8] Functional Genomics Facility, University of Colorado Anschutz Medical Campus, Aurora, Colorado, USA. [9] Department of Molecular, Cellular and Developmental Biology, University of Colorado Boulder, Boulder, Colorado, USA. *email: kelly.d.sullivan@cuanschutz.edu; joaquin.espinosa@cuanschutz.edu

Down syndrome (DS) is caused by triplication of chromosome 21 (chr21), which occurs in ~1 in 700 live births, representing the most common chromosomal abnormality in the human population[1]. Trisomy 21 (T21) impacts multiple organ systems during development and causes an altered disease spectrum in people with DS, significantly increasing their risk of developing Alzheimer's disease (AD)[2], leukemias[3], some respiratory infections[4], and numerous autoimmune conditions[1,5–8], while protecting them from solid malignancies[3,9]. However, DS-associated phenotypes are highly variable among individuals with DS, even for conserved phenotypes such as cognitive impairment and predisposition to early-onset AD[10,11]. Therefore, a deeper understanding of the mechanisms driving such phenotypic variation could illuminate not only mechanisms of pathogenesis, but also opportunities for diagnostics and therapeutic strategies to serve this population. Furthermore, these mechanistic insights could also benefit larger numbers of individuals in the typical population affected by the medical conditions that are modulated, either positively or negatively, by T21.

In order to identify molecular pathways that could contribute to both the altered disease spectrum experienced by the population with DS and the inter-individual phenotypic variation, our group previously analyzed transcriptomes[12] and circulating proteomes[13] from several cohorts of individuals with DS compared with euploid individuals (D21, controls). These efforts revealed that multiple cell types from people with DS show transcriptional signatures indicative of constitutive activation of the interferon (IFN) response[12], which could be explained by the fact that four of the six IFN receptors (IFNRs) are encoded on chr21: the two Type I IFNR subunits (IFNAR1, IFNAR2), one of the Type II IFNRs (IFNGR2), and IL10RB, which serves both as a Type III IFNR subunit and a subunit of the receptors for interleukin (IL)-10, IL-22, and IL-26[14]. Furthermore, plasma proteomics analyses demonstrated that people with DS show signs of chronic autoinflammation[13], including elevated levels of potent inflammatory cytokines with known ties to IFN signaling (e.g., IL-6, tumor necrosis factor (TNF-α), MCP-1). However, it remains to be defined how this obvious immune dysregulation contributes to DS-associated phenotypes, including the diverse immune and neurological manifestations of T21. Recent advances in the field of immunometabolism have highlighted a critical role for circulating levels of certain metabolites in the regulation of immune function and inflammation[15]. To gain further insight into this area, we investigated metabolic changes caused by T21 in the plasma and cerebrospinal fluid (CSF) of people with DS.

Here we report the results of a large plasma metabolomics study of individuals with DS from multiple cohorts. Initially, we performed an exploratory untargeted analysis of two cohorts via ultra-high-pressure liquid chromatography coupled to high-resolution mass spectrometry (UHPLC-HRMS) technology. The most significantly dysregulated metabolite identified was quinolinic acid (QA), a tryptophan (TRP) catabolite with well-characterized neurotoxicity, which has been repeatedly associated with diverse neurological disorders, and which is produced by activation of the kynurenine (KYN) pathway (KP). Indeed, people with DS have elevated QA, KYN, and KYN/TRP ratio, all established markers of KP activation. We then validated these findings using absolute quantification methods in additional cohorts, in which we confirmed the upregulation of KYN and QA in both the plasma and CSF of people with DS. We found that circulating immune cells of individuals with T21 show significant overexpression of indoleamine 2,3-dioxygenase 1 (IDO1), the rate-limiting enzyme in the KP, and a known IFN-stimulated gene (ISG) induced by all three types of IFN signaling. Using cell-based metabolic assays, we demonstrated that IFN-α stimulation leads to super-induction of IDO1 and hyperactivation of the KP

in cells with T21, and this requires overexpression of both Type I IFNR subunits encoded on chr21. Furthermore, circulating levels of key inflammatory cytokines, including IP-10, IL-10, and TNF-α, are positively correlated with KP activation in people with DS. Finally, we show that a mouse model of DS carrying triplication of the IFNR gene cluster, Dp(16)1/Yey, shows increased levels of KYN relative to wild-type (WT) littermates. Given the well-established neurotoxicity of QA, the association of KP dysregulation with a wide range of neurological conditions, as well as the immunosuppressive function of KYN, our results point to KP activation as a potential contributing factor to neurological and immunological phenotypes in DS.

## Results

**Trisomy 21 induces the kynurenine pathway.** To identify metabolic pathways that are consistently dysregulated by T21, we collected plasma samples from two fully independent cohorts: the Translational Nexus Clinical Data Registry and Biobank (referred hereto as Cohort 1) and the Crnic Institute's Human Trisome Project (www.trisome.org, NCT02864108, referred hereto as Cohort 2). Cohort 1 included 49 individuals with T21 and 49 euploid (D21) individuals, whereas Cohort 2 consisted of 26 T21 and 41 D21 (see demographics information in Supplementary Data 1). Although Cohort 1 was balanced for karyotype but imbalanced for age, Cohort 2 was more age-balanced, with a bias towards females in both groups. When the two cohorts are combined, age, sex, and karyotype are more evenly distributed, which prompted us to analyze the cohorts both individually and in combination. It should be noted that our protocol excluded participants with recent or active infections.

To determine metabolite abundance, we performed an exploratory UHPLC-HRMS-based metabolomics on plasma samples from each cohort. The combined analysis workflow is outlined in Supplementary Fig. 1a with details available in the Methods. This approach led to the identification of 91 Kyoto Encyclopedia of Genes and Genomes (KEGG)-annotated metabolites detected in both cohorts in >90% of the samples from at least one karyotype. To identify the metabolites that were consistently differentially abundant between people with and without DS, we applied a robust linear model-fitting approach that accounted for age and sex[16] to data from each of the cohorts (Supplementary Data 2–4). We observed both differences and overlaps in the differential metabolite profiles when analyzing Cohorts 1 and 2 individually. However, due to the relative nature of the quantification determined by exploratory UHPLC-HRMS, we observed an overall difference in the distributions of metabolite abundances between cohorts (Supplementary Fig. 1b). Therefore, to mitigate imbalances within the separate cohorts and to increase statistical power, we completed a combined cohort analysis by modeling age and sex covariates, along with fitting each cohort as a batch effect. Using this linear model fit, the distributions of metabolite abundance overlap and are thus comparable (Supplementary Fig. 1c).

When the combined total of 165 samples in the two cohorts were analyzed for the 91 metabolites using this linear model fit accounting for age, sex, and cohort, we identified 29 metabolites with relative levels that were significantly different according to a false discovery rate (FDR)-adjusted p-value on a Student's t-test of 0.05[17] (Fig. 1a and Supplementary Data 4). Consistent with earlier reports[18,19], L-serine and L-tyrosine were found to be depleted in people with DS (Fig. 1a and Supplementary Fig. 1d). Depletion of L-homocysteine (Fig. 1a and Supplementary Fig. 1d), which has also been previously observed in people with DS[20], could be attributed to the localization of the cystathionine-β-synthase gene (CBS) on chr21. Among the upregulated

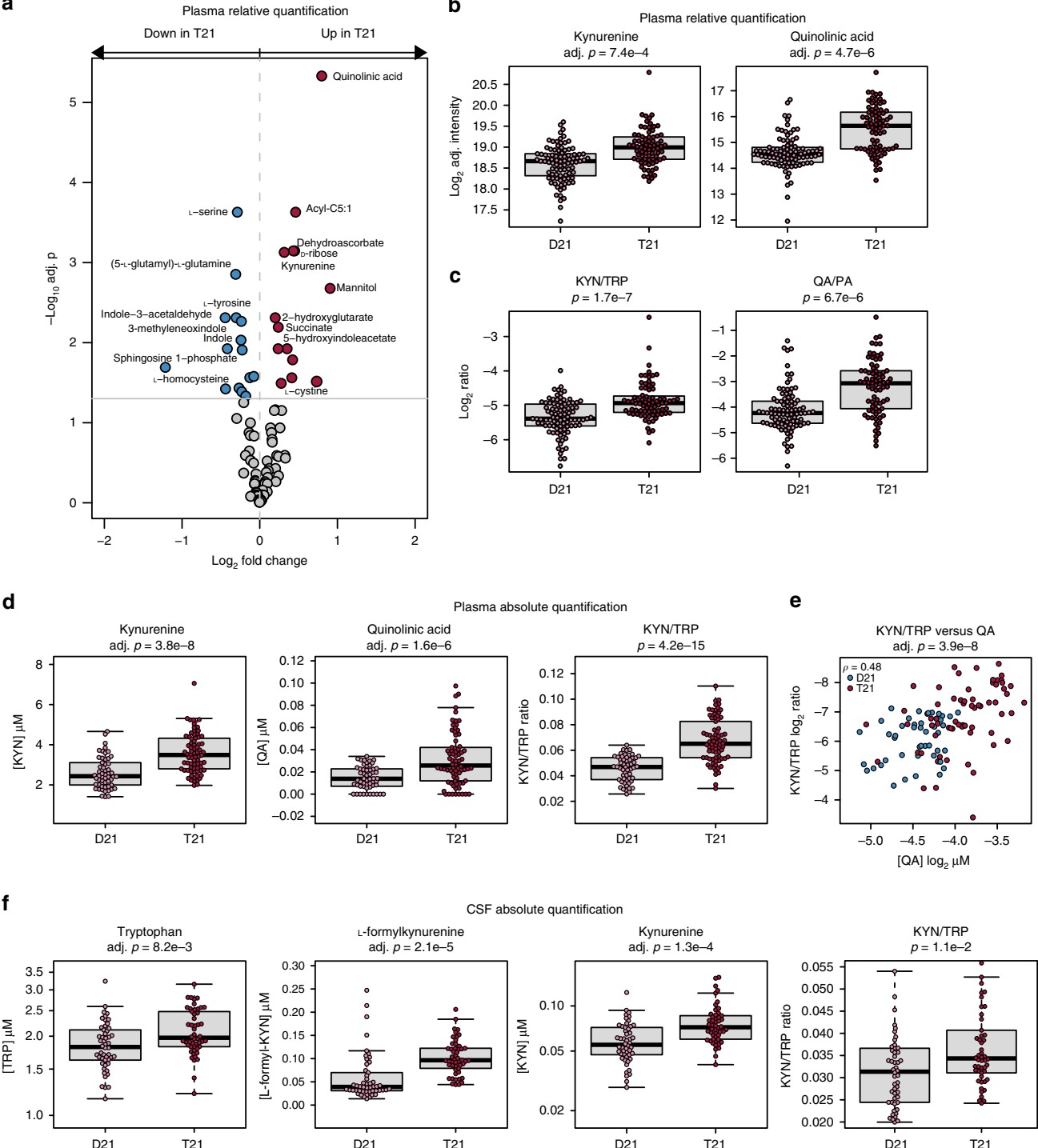

**Fig. 1** Trisomy 21 induces the kynurenine pathway. **a** Differentially abundant metabolites in plasma samples from individuals with T21 were identified using a linear model adjusting for age, sex, and cohort. Metabolites that increased in the T21 group are shown in red and metabolites that decreased in the T21 group are blue. The horizontal gray line represents an adjusted $p$-value of 0.05. $n = 165$ independent samples, 75 with T21. **b** Boxplots showing the relative quantification in $\log_2$-adjusted intensity for significantly differentially abundant metabolites. $p$-values were calculated using the linear model from **a** and the FDR method for multiple testing correction. **c** Boxplots showing $\log_2$-adjusted ratios as indicated. $p$-values were calculated using an unmoderated $t$-test on the linear model in **a**. **d** Boxplots showing concentrations of kynurenine and quinolinic acid, as well as the ratio of kynurenine to tryptophan calculated from the absolute quantification of these metabolites in Cohort 3, $n = 124$ independent samples, 72 with T21. **e** Scatter plots comparing kynurenine/tryptophan ratios with concentrations of quinolinic acid from Cohort 3. Spearman's rho ($\rho$) and FDR-corrected $p$-values are indicated. **f** Boxplots showing concentrations of tryptophan, L-formylkynurenine, and kynurenine, as well as the ratio of kynurenine to tryptophan calculated from the absolute quantification of these metabolites in the CSF of Cohort 4, $n = 100$ independent samples, 50 with T21. All boxplots show median, 25th, and 75th percentile values. Error bars are 1.5 times the interquartile range (IQR) or the maximum data point if < 1.5 IQR

metabolites, we observed succinate and 2-hydroxyglutarate, two metabolites related to the Krebs cycle (Fig. 1a and Supplementary Fig. 1d). Elevated succinate levels have been previously reported in people with DS and were interpreted as a sign of mitochondrial dysfunction[19], but succinate upregulation is also an established marker of hypoxia and inflammation[21,22]. Other upregulated metabolites included dehydroascorbate and mannitol (Fig. 1a and Supplementary Fig. 1d), molecules involved in redox metabolism[23,24]. The most significantly dysregulated metabolite in this analysis was QA, an intermediate in the TRP catabolic pathway that ultimately leads to synthesis of nicotinamide adenine dinucleotide (NAD$^+$). In fact, of the 11 measured metabolites in the TRP pathway annotated by KEGG, we found five to be differentially abundant in people with DS: KYN, QA, and 5-hydroxyindoleacetate are upregulated, whereas indole and indole-3-acetaldehyde are downregulated (Fig. 1a, b and Supplementary Fig. 2a).

In vertebrate cells, there are three major catabolic pathways for TRP, leading to the production of serotonin, NAD$^+$, or indole (Supplementary Fig. 2a). Conversion of TRP to KYN represents a major pathway for catabolism of ingested TRP, eventually leading to NAD$^+$ production, and accounting for the removal of up to 99% of TRP not used in protein synthesis[25]. Remarkably, KYN itself and the neurotoxic downstream product QA are significantly elevated in people with DS (Fig. 1a, b). In contrast, picolinic acid (PA), a neuroprotective derivative of KYN, is not different between the two groups (Supplementary Fig. 2a). Accordingly, the KYN/TRP and QA/PA ratios are both significantly higher in people with DS (Fig. 1c).

Having revealed KP dysregulation in people with T21, we created a third cohort of 128 plasma samples (Cohort 3, 74 T21, 54 D21; Supplementary Data 1), which partially overlapped with, and expanded upon, Cohort 2 to validate these findings in a quantitative manner comparable to other studies. For 124 samples in Cohort 3, we employed a modified extraction protocol designed to improve chromatographic separation, peak resolution, and signal intensity for TRP pathway metabolites and spiked in stable isotopically labeled TRP and KYN, allowing for the determination of the absolute quantities of TRP, KYN, and several other metabolites in the pathway. This targeted quantitative analysis of Cohort 3 confirmed elevated levels of KYN, QA, and KYN/TRP ratios in people with T21 (Fig. 1d and Supplementary Data 5 and 6). KYN levels for D21 control samples had a median value of 2.43 µM, within the range observed in other reports (values ranging from 712 nM to 2.95 µM[26–29]; Supplementary Data 7). Importantly, the median concentration of KYN in the plasma of people with DS is 3.49 µM, higher than reported for AD[26], depression[27], and lupus[28], and similar to levels seen in patients with fatal cardiac arrest[29]. QA levels were lower in both our D21 and T21 cohorts relative to other studies[26–29]; however, the elevation in QA levels seen in people with T21 represents the largest median fold change associated with a medical condition among these studies (~1.9-fold; Supplementary Data 6 and 7). TRP levels for both D21 and T21 were in line with other studies and otherwise unremarkable. Spearman's rank correlation analysis between the KYN/TRP ratios and QA levels from all individuals in Cohort 3 demonstrated a highly significant correlation between these two parameters (Fig. 1e and Supplementary Data 8).

Finally, to assess whether the KP is dysregulated in the central nervous system (CNS), we performed metabolomics analysis in CSF samples from a fourth cohort (Cohort 4, 50 D21, 50 T21; Supplementary Data 1), using the same quantitative method as for Cohort 3. This experiment confirmed KP dysregulation in T21 as evidenced by elevated levels of TRP, L-Formylkynurenine, and KYN itself (Fig. 1f and Supplementary Data 9 and 10).

Interestingly, despite the increased levels of TRP, the KYN/TRP ratio is significantly elevated in the CSF of people with T21, consistent with activation of the KP (Fig. 1f). Of note, although QA levels were below the limit of detection in many samples (36/50 D21, 23/50 T21), it was significantly elevated in the T21 group (Supplementary Fig. 1e).

Altogether, these results indicate that a major metabolic impact of T21 is dysregulation of TRP catabolism towards increased activity in the KP. Importantly, activation of the KP has been implicated in myriad neurological and neurodegenerative disorders, and elevation in the KYN/TRP and QA/PA ratios have been linked to the development of neurological conditions such as AD[30–33], Huntington's disease[34], AIDS/HIV-associated neurocognitive disorder[35], Parkinson's disease[36], amyotrophic lateral sclerosis[37,38], and multiple sclerosis[39]. In addition, KYN has potent immunosuppressive functions[40]. Therefore, we focused our efforts on elucidating the mechanisms driving activation of the KP in DS.

### KP activation correlates with *IDO1* and IFN-related cytokines.
To investigate potential mechanisms driving the elevated levels of KYN and QA in people with DS, we performed transcriptome analysis of circulating white blood cells (WBCs) in 19 adult individuals, 10 of them with T21 (see Supplementary Data 1, Cohort 5). We interrogated this dataset to assess possible alterations in the expression levels of various enzymes catalyzing different reactions in the KP (see Supplementary Fig. 2a, b). This exercise revealed a significant increase in the expression of *IDO1*, the enzyme that catalyzes the rate-limiting step in the KP, in people with DS (Fig. 2a and Supplementary Data 11). The only other significant change observed was a decrease in KYN 3-monooxygenase, the enzyme that converts KYN into 3-hydroxykynurenine (Supplementary Fig. 2a–c). Given that *IDO1* is a well-characterized ISG, known to be stimulated by all three types of IFN signaling[41–44], this systemic *IDO1* overexpression could be explained by our earlier finding that different immune cell types from people with DS show consistent hyperactivation of the IFN response[12]. Thus, these results indicate that the observed activation of the KP in people with DS could be explained simply by higher levels of IFN signaling and *IDO1* overexpression.

We recently reported the results of a large plasma proteomics study of people with DS[13], which revealed that T21 causes changes in the circulating proteome indicative of chronic autoinflammation, including elevated levels of many cytokines acting downstream of IFN signaling. Therefore, we tested whether there was a correlation between levels of inflammatory cytokines and KP activation by using a Mesoscale Discovery (MSD) assay to measure a panel of 55 cytokines in the plasma samples from the 128 participants in Cohort 3. This analysis demonstrated significant upregulation of many potent cytokines in people with DS, including IL-10, IP-10, IL-6, IL-22, TNF-α, MCP-1, CRP, and several others (Fig. 2b and Supplementary Data 12 and 13)[13]. Interestingly, all four IFN ligands measured (IFN-α2a, IFN-β, IFN-γ, and IFN-λ1/IL-29) were elevated in the population with T21, although only IFN-α2a was significant at an FDR-adjusted *p*-value on a Kolmogorov–Smirnov test of 0.05 (Supplementary Fig. 3 and Supplementary Data 12 and 13). Comparison of our MSD data with a comparable study of plasma cytokines in systemic lupus erythematosus, an IFN-driven autoimmune disorder, revealed that all 11 cytokines measured in both studies were significantly elevated in both conditions[45], supporting the notion that the inflammatory profile of DS could be associated with increased IFN signaling (Supplementary Data 14).

To investigate the relationship between inflammatory cytokines and KP dysregulation, we calculated Spearman's rank correlations for each cytokine measured and the KYN/TRP ratio, which

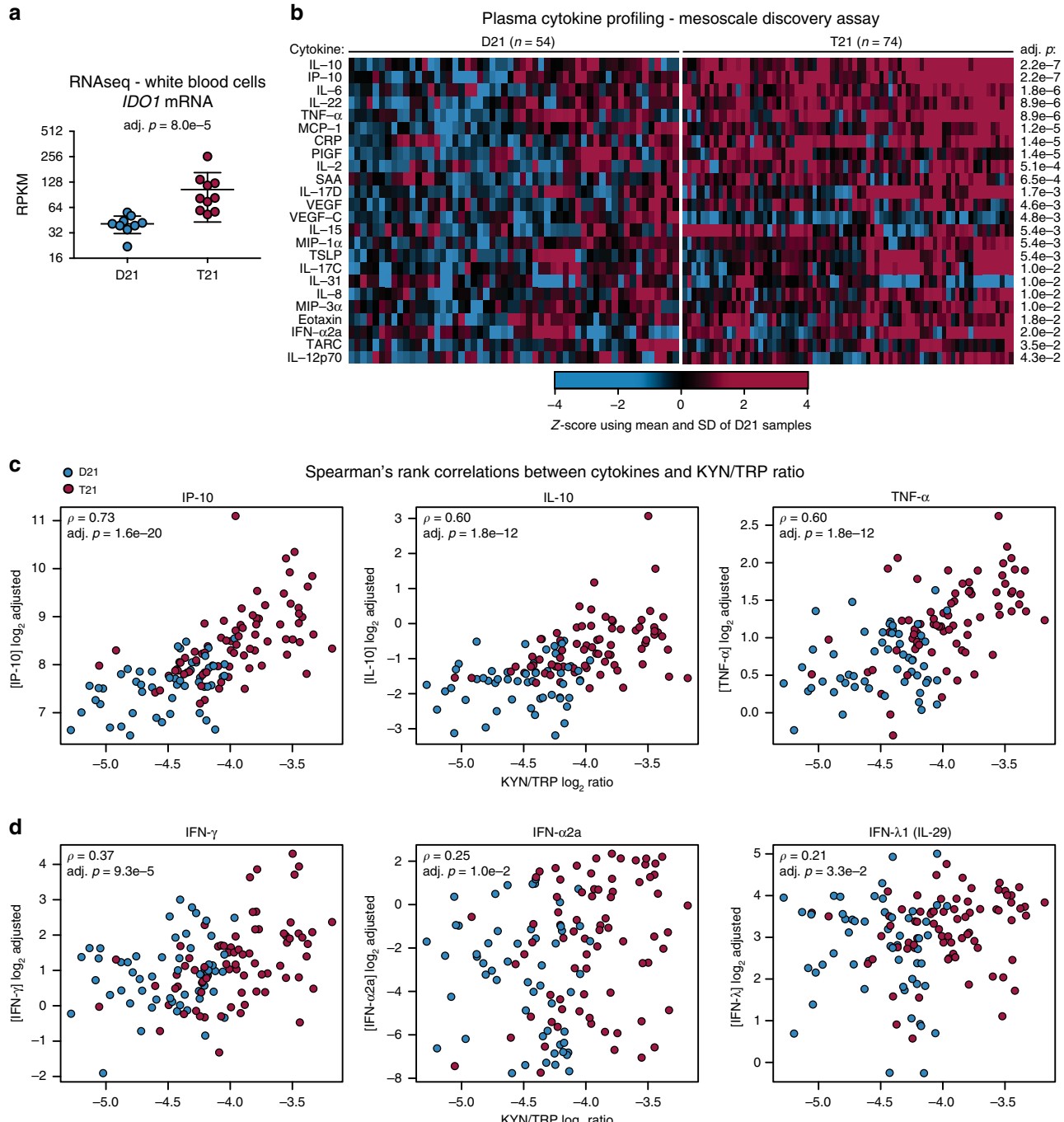

**Fig. 2** KP activation correlates with *IDO1* and IFN-related cytokines. **a** Scatter plot showing mRNA expression of *IDO1* in white blood cells from controls (D21) and individuals with T21. Statistical significance was calculated using DESeq2. mRNA expression values are displayed in reads per kilobase per million (RPKM). Bars represent median, 25th, and 75th percentile values. $n = 19$ independent samples 10 with T21. **b** Heat map showing significantly different cytokines identified in Mesoscale Discovery Assay in Cohort 3. Data from both the D21 and T21 samples were z-scored using the mean and SD from the D21 group. Significant differences were assessed using the Kolmogorov–Smirnov test and an FDR-adjusted *p*-value threshold of 0.05. **c** Scatter plots comparing kynurenine to tryptophan ratios to levels of the three most significantly correlated cytokines, IP-10, IL-10, and TNF-α. **d** Scatter plots comparing kynurenine to tryptophan ratios to levels of three significantly correlated interferons: IFN-γ, IFN-α2a, and IFN-λ (IL-29). Spearman's rho (ρ) and FDR-corrected *p*-values are indicated for each pair. Panels **b**–**d** measured Cohort 3, $n = 128$ independent samples, 74 with T21 for panel **b**, $n = 124$ samples, 72 with T21 for panels **c** and **d**

identified 31 significant correlations (Supplementary Data 15). At the top of this list were three cytokines known to be induced by IFN signaling, the proinflammatory molecules IP-10 (IFN-γ-induced protein 10, CXCL10) and TNF-α, as well as the anti-inflammatory cytokine IL-10 (Fig. 2c and Supplementary Data 15). Furthermore, three IFN ligands were positively

correlated with the KYN/TRP ratio, albeit to a lesser degree: IFN-γ, IFN-α2a, and IFN-λ1 (Fig. 2d and Supplementary Data 15), each of which have been shown to induce the expression of *IDO1*[41–44].

Taken together, these data indicate that increased KYN levels are associated with *IDO1* overexpression and increased levels of

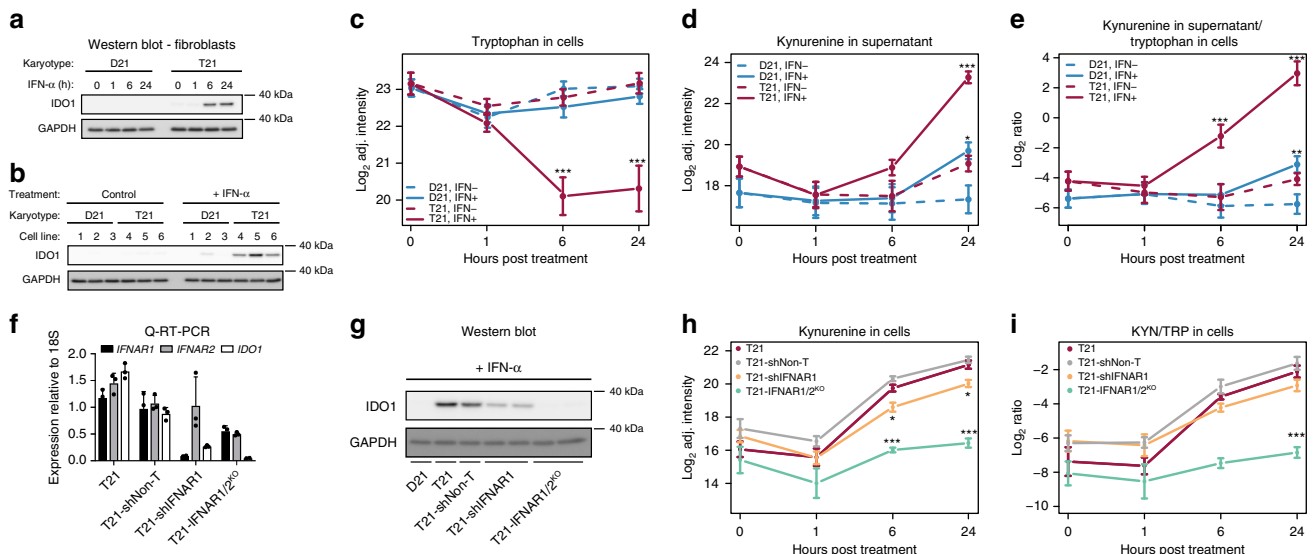

**Fig. 3** T21 sensitizes cells to KP induction via IFNR gene dosage. **a** Western blottings demonstrating the upregulation of IDO1 over a 24 h time course in an age- and gender-matched pair of fibroblasts with (T21) or without (D21) trisomy 21. **b** Western blot analysis confirming IDO1 super-induction in multiple T21 fibroblasts after 24 h of IFN-α treatment. 1–3 are cell lines from controls, 4–6 are cell lines from individuals with trisomy 21. **c** Metabolic tracing experiment using isotopolog-labeled ($^{13}C_{11}^{15}N_2$) tryptophan in fibroblast cell lines ($n = 6$, 3 with T21). Levels of isotopolog-labeled tryptophan in D21 and T21 cell lysates, with or without IFN-α treatment. **d** Levels of isotopolog-labeled kynurenine in D21 and T21 supernatant, with or without IFN-α treatment. **e** Ratio of isotopolog-labeled kynurenine in supernatant to isotopolog-labeled tryptophan levels in cells in D21 and T21 supernatant, with or without IFN-α treatment. At each timepoint, the $p$-value between the untreated (IFN-) and IFN-treated (IFN+) samples was calculated using a two-tailed Student's $t$-test. **f** Q-RT-PCR showing relative mRNA levels of *IFNAR1*, *IFNAR2*, and *IDO1*, in the indicated cell lines. T21-shNon-T indicates cell line expressing a non-targeting shRNA. T21-shIFNAR1 indicates cell line expressing a shRNA targeting IFNAR1. T21-IFNAR1/2KO indicates a cell population transduced with guide RNAs targeting the first exons of *IFNAR1* and *IFNAR2*. mRNA expression is expressed relative to 18S ribosomal RNA. **g** Western blotting showing effects of modulating *IFNAR* expression as in **f** on induction of IDO1 protein after 6 h of IFN-α stimulation. **h** Levels of isotopolog-labeled kynurenine in the indicated cell lines during a 24 h time course of IFN-α treatment. **i** Ratio of isotopolog-labeled kynurenine to isotopolog-labeled tryptophan levels in the indicated cell lines during a 24 h time course of IFN-α treatment. At each timepoint, the $p$-value between the parental T21 cell line and each T21 cell line with modified IFNAR expression was calculated using a two-tailed Student's $t$-test. Data shown in all panels are mean ± SEM, $*p < 0.05$, $**p < 0.01$, $*** p < 0.001$. All experiments were performed in triplicate for a total of $n = 18$

specific inflammatory markers in vivo, consistent with constitutive IFN hyperactivity and immune dysregulation in DS.

**T21 sensitizes cells to KP induction via *IFNR* gene dosage.** Given the wealth of potential mechanisms that could alter TRP catabolism in individuals with DS, including differences in medical histories, existing co-morbidities, dietary regimes, and medication intake, we asked whether KP dysregulation could be observed at the cellular level. Towards this end, we employed cell-based metabolic tracing experiments using stable isotope-labeled ($^{13}C_{11}^{15}N_2$) TRP on a panel of age- and sex-matched skin fibroblasts derived from individuals with and without T21, both before and after stimulation with recombinant human IFN-α2a. Matched western blot analysis showed that T21 fibroblasts display much stronger induction of IDO1 protein expression relative to D21 cells (Fig. 3a, b). Remarkably, IFN-α stimulation produced a significant, time-dependent depletion of the isotopic TRP in T21 cells, but not supernatant (Fig. 3c, Supplementary Fig. 4a and Supplementary Data 16), concurrent with a significant increase in KYN levels in the supernatant, but not in cells (Fig. 3d and Supplementary Fig. 4b). Accordingly, the KYN/TRP ratios were most significantly elevated upon IFN-α stimulation in T21 cell cultures, consistent with more pronounced consumption of intracellular TRP and subsequent secretion of elevated levels of KYN in cells of people with DS (Fig. 3e and Supplementary Fig. 4c, d).

Next, we tested whether the observed super-induction of IDO1 and the KP by IFN-α in T21 cells is due to the extra copy of the

Type I *IFNR*s encoded on chr21 (*IFNAR1*, *IFNAR2*), using both short hairpin RNA (shRNA)-mediated knockdown and CRISPR-based knockout approaches. First, we stably knocked down *IFNAR1* expression in one of the T21 cell lines by >90% with a specific shRNA that did not affect *IFNAR2* expression (T21-shIFNAR1; Fig. 3f). Knocking down *IFNAR1* resulted in a reproducible decrease in IFN-α-induced IDO1 expression at both the mRNA and protein levels (Fig. 3f, g). Independently, we also co-transduced two independent guide RNAs (gRNAs) targeting the first exons of *IFNAR1* and *IFNAR2*, which created a population of T21 fibroblasts expressing ~50% of both the *IFNAR1* and *IFNAR2* mRNAs relative to the parental cell line (T21-*IFNAR1/2*KO; Fig. 3f). Notably, such reduction in the expression of both Type I IFNRs completely abolished over-expression of *IDO1* mRNA and protein in response to IFN-α stimulation, bringing it down to levels comparable to those observed in the D21 cell line (Fig. 3f, g). We then repeated the metabolic tracing experiment with $^{13}C_{11}^{15}N_2$-labeled TRP on this panel of T21 cell lines with reduced *IFNR* expression during a 24 h time course of IFN-α treatment. Remarkably, the T21-*IFNAR1/2*KO cell line showed significantly reduced levels of KYN in both cells and supernatants, as well as significantly reduced KYN/TRP ratios (Fig. 3h, i and Supplementary Fig. 4e–f). Furthermore, the T21-shIFNAR1 cell line showing intermediate IDO1 expression displayed a partial rescue of elevated KYN levels in cells, but not in supernatants (Fig. 3h and Supplementary Fig. 4f).

Altogether, these results indicate that IDO1 induction by IFN-α is highly sensitivity to *IFNR* dosage, providing a mechanism by which T21 could sensitize cells to IFN-dependent super-activation

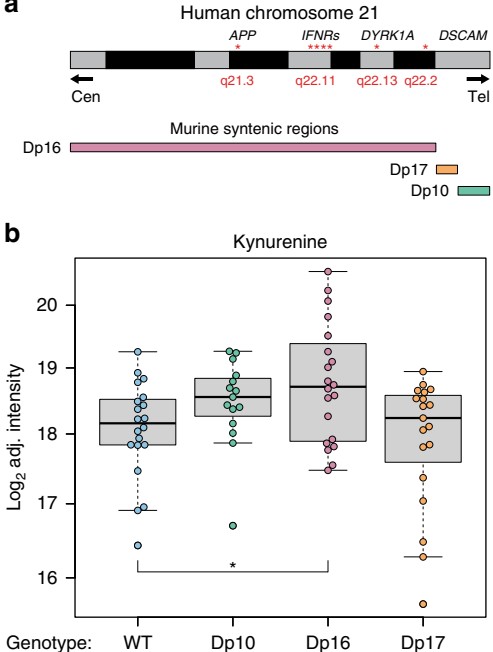

**Fig. 4** Tryptophan catabolism is disrupted in the *Dp16* model of DS.
**a** Schematic of mouse models tested showing the localization key genes on chromosome 21, including four of the six IFN receptors in Dp16 mice (*Ifnar1, Ifnar2, Ifngr2,* and *Il10rb*). **b** Box and whisker plots showing the relative quantification in log$_2$ adjusted intensity of plasma kynurenine in the indicated mouse model. *p*-values were calculated using a two-tailed Student's *t*-test (**p* < 0.05). Boxplot shows median, 25th, and 75th percentile values. Error bars are 1.5 times the interquartile range (IQR) or the maximum data point if <1.5 IQR. N's for each strain were WT−20, Dp10−15, Dp16−20, Dp17−19

of the KP. These results also demonstrate that KP dysregulation is observed at the cellular level, indicating that the elevated levels of KYN and QA in the circulation of people with DS is likely driven by the trisomy, rather than other variables.

**Tryptophan catabolism is disrupted in the *Dp16* model of DS**.
To test the impact of various groups of genes encoded on human chr21 on TRP metabolism, we employed three cytogenetically distinct mouse strains carrying segmental duplications syntenic to human chr21[46]: *Dp(10)1/Yey* (Dp10), *Dp(16)1/Yey* (Dp16), and *Dp(17)1/Yey* (Dp17), which have triplication of ~35 genes from mouse chr10, ~120 genes from chr16, and ~20 genes from chr17, respectively, and compared them with WT C5Bl/6 mice (Fig. 4a). We prepared plasma samples from age- and sex-matched cohorts comprising these strains and measured KYN using UHPLC-HRMS. Interestingly, despite clear variability among individual mice, the only strain with significantly increased plasma KYN levels relative to WT mice was the Dp16 model, which carries the largest segmental duplication, including the four IFNRs encoded on human chr21: *Ifnar1, Ifnar2, Ifngr2,* and *Il10rb* (Fig. 4a, b and Supplementary Data 17). Furthermore, we and others have previously shown that not only are the IFNRs overexpressed in this model, but downstream IFN signaling is chronically activated on numerous tissues including blood and brain[12,47]. Of note, a recent report demonstrated elevation of KYN levels in brain tissue of the Ts65Dn mouse model of DS, which also carries triplication of the *Ifnr* gene cluster[48]. These results are consistent with a genetic basis for the observed dysregulation of the KP and further implicate triplication of the IFNR locus as a key contributor. Finally, it is notable that the Dp16 mouse strain has the most

profound neurological impairment relative to Dp10 and Dp17. For example, Dp16 mice have demonstrated defects in spatial learning and memory, context-associated learning, and decreased hippocampal long-term potentiation, not present in Dp10 or Dp17 mice[46].

## Discussion

The developmental and clinical impacts of T21 in the population with DS are diverse and highly variable, with many co-morbidities showing increased prevalence in this population relative to typical people. Despite many research efforts in this area, little is known about the molecular pathways that drive the various co-morbidities common in DS. Clearly, identification of these pathways could facilitate the design of diagnostic and therapeutic strategies to serve this at-risk population via precision medicine approaches. An obvious impact of these approaches would be in the management of diverse neurological conditions showing increased prevalence in this population, including, but not restricted to, early-onset AD[2,11], epilepsy[49], depression[50], and autism[51]. Another key area of interest is in the management of immune-related disorders. On one hand, people with DS show increased incidence of autoimmune disorders such as autoimmune hypothyroidism[52], celiac disease[8], and myriad autoimmune skin conditions[53], indicating a hyperactive immune system. Conversely, people with DS show increased risk to specific bacterial infections, with bacterial pneumonia being a leading cause of death[4,54]. Our results point to dysregulation of the KP as a potential contributing factor to the development of some neurological and immunological phenotypes in people with DS, with obvious diagnostics and therapeutic implications as described below.

People with DS constitute the largest population with a strong predisposition to AD, which is attributed to the fact that the *APP* gene is located on chr21[2]. By age 40 years, virtually all individuals with DS present the brain pathology of AD; however, the age of dementia diagnoses is highly variable, with some individuals being diagnosed in their 30s, whereas others remain dementia-free in their 70s[11]. Therefore, the population with T21 provides a unique opportunity to identify mechanisms that modulate the impact of amyloidosis to either accelerate or delay AD development. Interestingly, the KP has been consistently implicated in AD progression. Multiple studies have reported elevated levels of metabolites in the KP in people with AD and, in many of these studies, KP dysregulation has been positively correlated with AD progression (reviewed in ref. [55]). Furthermore, IDO1 inhibition reversed AD pathology in a mouse model of AD[56]. Therefore, it is possible to hypothesize that individuals with DS showing the strongest degree of dysregulation in the IFN→IDO1→KYN axis may display earlier onset or faster progression of AD. Clearly, testing this hypothesis will require a significant effort, including a multi-dimensional, longitudinal analysis of IFN signaling, KP dysregulation, and various metrics of the AD pathological cascade in people with DS.

Beyond AD, it is possible that KP dysregulation is a contributor to other neuropathologies more prevalent in DS, such as seizure disorders, depression, and autism[49–51]. KYN and QA were shown to promote seizures in mice, rats, and frogs, and QA was found to be elevated in mouse models of epilepsy, leading to the characterization of QA as an "endogenous convulsant" with a pathogenic role in seizure disorders (reviewed in ref. [57]). Regarding depression, activation of the KP shunts TRP away from serotonin synthesis, causing serotonin depletion, which led Lapin and others to propose the "kynurenine hypothesis" of depression[58]. Although we did not observe a significant depletion of serotonin in our research participants with DS (Supplementary Fig. 2a), future studies could investigate whether individuals with

DS diagnosed with depression show increased activation of the KP. Notably, it has been amply documented that therapeutic administration of IFN-α can induce depression, leading Wichers et al.[43] to propose that IDO1 induction (and consequent neuro-toxicity) was the principal pathophysiological mechanism. Regarding autism, independent cohort studies have documented dysregulation of KYN metabolism in patients diagnosed with autism spectrum disorder[59,60], as demonstrated by increased KYN/kynurenic acid ratios[60] and elevated levels of QA[59]. KP dysregulation could also potentially contribute to the cognitive deficits associated with DS. The acute effects of KP metabolites, QA in particular, on cognitive function are well-documented[61,62]. QA is a potent neurotoxin that acts as an excitotoxic agonist of NMDA receptors, which could explain the beneficial effects of memantine, an NMDA antagonist that protects from QA-mediated neurotoxicity, in animal models of DS[63–65]. Other bioactive metabolites in the KP that were not measured in our study, such as formic acid, could also contribute to developmental and clinical impacts of T21. In fact, formate was recently reported to be elevated in plasma of people with DS[19]. Lastly, TRP cata-bolites produced by the gut flora such as indole and indole-3-acetaldehyde, both of which are decreased in individuals with T21 (Supplementary Fig. 2a), were shown to suppress inflammation in the CNS via activation of the aryl hydrocarbon receptor in astrocytes[66], demonstrating additional roles for TRP catabolites outside of the KP in immune regulation and neurological function.

It is important to note that the KP plays major roles in immune control. KYN is a potent immunosuppressive metabolite pro-moting "immune privilege" in tissues such as the brain, testes, and gut, but which is also exploited by some tumor types to evade immune surveillance[40]. KYN has complex effects on immune cells, leading to suppression of effector CD4 and CD8 T cells, while also inducing regulatory T cells (Tregs, reviewed in ref. [67]) and it may represent an immunosuppressive negative feedback mechanism downstream of exacerbated IFN signaling. In fact, Tregs have been shown to be much elevated in the circulation of people with DS[68], which could contribute to both autoimmune conditions and poor resolution of inflammatory processes. Given that people with DS are highly predisposed to myriad auto-immune disorders[1,5–8] and increased mortality due to pneumonia lung infections[54], our results justify future investigations to define the role of KP dysregulation in the function of diverse immune cell types in particular and the development of common co-morbidities in DS in general.

## Methods

**Human cohorts and sample collection.** All human subjects in this study were consented according to Colorado Multiple Institutional Review Board (COMIRB)- and Sant Pau Ethic Committee-approved protocols. Written informed consent was obtained from parents or guardians of participants under the age of 18 years and assent was obtained from participants over the age of 7 years, who were cognitively able to assent. Deidentified plasma samples for Cohort 1 were obtained from the Translational Nexus Clinical Data Registry and Biobank (University of Colorado Anschutz Medical Campus, COMIRB #08–1276). Additional plasma and WBC samples for Cohorts 2 and 3 were obtained through the Crnic Institute's Human Trisome Project (University of Colorado Anschutz Medical Campus, COMIRB #15–2170, www.trisome.org). Cohort 4, including T21 CSF samples, was obtained through the Down Alzheimer Barcelona Neuroimaging Initiative run at Barcelona Down Medical Center. Control CSF samples were obtained from the Sant Pau Initiative on Neurodegeneration cohort. All cohorts are described in Supplementary Data 1. Plasma was collected in Vacutainer tubes (EDTA—purple capped or Lithium heparin—light green capped) and stored at −80 °C. Participant medical history was collected by the respective biobanks. CSF samples were collected as previously described[69].

**Sample extraction for metabolomics.** Samples were extracted in an ice-cold methanol:acetonitrile:water (5:3:2 *v/v*) solution at the following ratios: for plasma analyses (human and mouse plasma), a volume of 20 μL was extracted in 480 μL of

the solution; for CSF analyses, a volume of 20 μL was extracted in 180 μL of the same ice-cold solution; and for cell-based experiments, two million cells were extracted in 1 mL of the same ice-cold lysis solution. Subsequently, these solutions were vortexed for 30 min at 4 °C. Insoluble proteins were pelleted by centrifugation (10 min at 4 °C and 12,000 × g) and supernatants were collected and stored at −80 °C until analysis.

For absolute quantification experiments, $^{13}C_{10}$ KYN and $^{15}N_2$ TRP (Cambridge Isotope Laboratories, Inc., Andover, MA) were added to the lysis solution prior to extraction at a final concentration of 2 μM and compounds of interest were monitored and quantified against these internal standards. For the quantitative analysis of KP metabolites, the extraction protocol also included a final additional step in which supernatants were further spun in a Speedvac until dry and resuspended in 0.1% formic acid in water. This approach improves chromatographic separation and signal intensity of metabolites in the KP, as previously validated in technical notes[70].

**UHPLC-HRMS conditions for metabolomics.** All UHPLC-HRMS metabolomics analyses were performed using a Vanquish UHPLC system coupled online to a high-resolution Q Exactive mass spectrometer (Thermo Fisher, Bremen, Germany). Samples were resolved over a Kinetex C18 column (2.1 × 150 mm, 1.7 μm; Phenomenex, Torrance, CA, USA) at 45 °C. A volume of 10 μl of sample extracts was injected into the UHPLC-MS. Each sample was injected and run four times with two different chromatographic and MS conditions as follows: (1) using a 5 min gradient at 450 μL/min from 5% to 95% B (A: water/0.1% formic acid; B: acetonitrile/0.1% formic acid) and the MS was operated in positive mode and (2) using a 5 min gradient at 450 μL/min from 5% to 95% B (A: 5% acetonitrile, 95% water/1 mM ammonium acetate; B:95% acetonitrile/5% water, 1 mM ammonium acetate) and the MS was operated in negative ion mode. The UHPLC system was coupled online with a Q Exactive (Thermo, San Jose, CA, USA) scanning in Full MS mode at 70,000 resolution in the 60–900 m/z range, 4 kV spray voltage, 15 sheath gas, and 5 auxiliary gas, operated in negative or positive ion mode (separate runs). Calibration was performed prior to analysis using the PierceTM Positive and Negative Ion Calibration Solutions (Thermo Fisher Scientific). Acquired data were converted from.raw to.mzXML file format using Raw-Converter[71]. To monitor possible technical variability, aliquots of each of the individual samples were combined to make technical replicates, which were run for both cohorts after every 15 samples. In addition, in each experiment, several lysis solution aliquots were run as blanks for artifact identification. Coefficients of variations (CV = SD/mean) were calculated for each metabolite across tech mixes and only metabolites with CVs < 20% were considered for this manuscript.

**Metabolomics data processing.** Metabolite assignments to KEGG compounds, isotopolog distributions, and correction for expected natural abundances of $^{13}C$ and $^{15}N$ isotopes, were performed using MAVEN (Princeton, NJ, USA) on the basis of accurate intact mass, high-resolution-based determination of chemical formulae, retention times against an in-house standard library of ~1000 compounds[72,73]. Relative quantification was performed in both positive and negative ion modes, according to the ionizability of each metabolite—as determined against an in-house standard library of ~1000 compounds, as validated in previous technical notes[74,75]. Peak areas were exported for further statistical analysis with R (R Foundation for Statistical Computing, Vienna, Austria). Data sets from exploratory, untargeted analyses were $\log_2$ transformed. Subsequent data preprocessing included filtering out metabolites with low-intensity values (per-metabolite threshold informed by blanks) that were detected in <90% of the T21 samples or <90% of the D21 samples, leaving 91 metabolites for further analyses. To perform differential expression analysis, the limma R package (v3.32.10) was used to fit a linear model to the data with age, sex, and cohort as covariates[16]. $\log_2$ fold change, p-value and adjusted p-value were calculated for the T21–D21 comparison using an unmoderated Student's t-test and the FDR method for multiple testing correction[17]. Adjusted p-values are shown in the volcano plots and reported on the boxplots. For displaying data per sample on the boxplots, we plotted the adjusted intensity values for each metabolite produced by the remo-veBatchEffects function in the limma R package using age, sex, and cohort as covariates. Stable isotope-labeled TRP and KYN from tracing experiments were monitored in positive ion mode. Isotopolog distributions in tracing experiments with $^{13}C_{11}$$^{15}N_2$-TRP were also manually analyzed with MAVEN (Princeton, NJ, USA).

**Calculation of absolute quantification data.** Calculation of absolute quantification for KP metabolites was performed using the following formula: [light] = (abundance light)/(abundance heavy)*[heavy] [dilution factor] where [light] = concentration of non-isotopic metabolite, (abundance light) = total area abundance of non-isotopic metabolite, (abundance heavy) = total area abundance of isotopic metabolite, and [heavy] = known concentration isotopic metabolite.

**RNA-seq sample prep.** Peripheral blood was collected in EDTA vacutainer tubes (BD, 366643) from ten individuals with T21 and nine D21 controls. Blood was centrifuged at 500 × g for 15 min to separate plasma, buffy coat, and red blood cells. WBCs were isolated from the buffy coat fraction by RBC lysis and washed with our

sorting buffer (see below) according to the manufacturer's instructions (BD, 555899). WBCs were cryopreserved in CryoStor CS10 Freezing Media (210102) at ~10 million cells/mL. WBCs were quickly thawed at 37 °C and immediately spiked into 9 mL of sorting media and centrifuged at $500 \times g$ for 5 min and the media removed. Cells were lysed in 600 μL RLT plus (Qiagen) and β-mercaptoethanol (BME) lysis buffer (10 μL BME:1 mL RLT plus). RNA was extracted using the AllPrep DNA/RNA/Protein Mini Kit according to the manufacturer's instructions (Qiagen, 80004). RNA quality was determined by Agilent 2200 TapeStation and quantified by Qubit (Life Technologies). Samples with RIN of 6.8 or greater and a minimum of 1 μg were sent to Novogene for stranded library prep and $2 \times 150$ paired-end sequencing.

**RNA-seq data analysis.** Analysis of library complexity and high per-base sequence quality across all reads (i.e., $q > 30$) was performed using FastQC (v0.11.5) software (Andrews 2010). Low-quality bases ($q < 10$) were trimmed from the 3′-end of reads and short reads (<30 nt after trimming), and adaptor sequences were removed using the fastqc-mcf (v1.05) tool from ea-utils. Common sources of sequence contamination such as mycoplasma, mitochondria, and ribosomal RNA were identified and removed using FASTQ Screen (v0.9.1). Reads were aligned to GRCh37/hg19 using TopHat2 (v2.1.1,–b2-sensitive–keep-fasta-order–no-coverage-search–max-multihits 10–library-type fr-firststrand). High-quality mapped reads (MAPQ > 10) were filtered with SAMtools (v1.5). Reads were sorted with Picard-tools (v2.9.4) (SortSAM) and duplicates marked (MarkDuplicates). Quality control of final reads was examined using RSeQC (v2.6.4). Gene-level counts were obtained using HTSeq (v0.8.0,–stranded = reverse –minaqual = 10 –type = exon –idattr = gene_id–mode = intersection-nonempty, GTF-ftp://igenome:G3nom3-s4u@ussd-ftp.illumina.com/Homo_sapiens/UCSC/hg19/ Homo_-sapiens_UCSC_hg19.tar.gz). Differential expression was determined using DESeq2 (v1.18.1) and R (v3.4.3).

**Cell culture.** Human fibroblasts cell lines GM08447, GM02036, GM05659, GM04616, GM02767, and AG05397 were described previously[12]. For the metabolic tracing experiment, cells were cultured in 5 mg/L $^{13}C_{11}{}^{15}N_2$ L-tryptophan (Sigma-Aldrich, 574597) for 72 h prior to stimulation with 10 ng/mL IFN-α2a (R&D Systems, 11101–2). Media was prepared by adding 120 μL of 10 mg/mL stock $^{13}C_{11}{}^{15}N_2$ L-tryptophan in 1× phosphate-buffered saline (PBS) to 240 mL of RPMI (Corning, 10–040) in 10% dialyzed fetal bovine serum (FBS) (Sigma, F0392) and 1× Anti-Anti (Gibco, 15240–062). Cells were treated with 10 ng/mL IFN-α2a or vehicle by adding 6 μL of 0.2 mg/mL IFN-α2a or 1× PBS to 120 mL of isotopolog-labeled media. Cell lysates and media were collected at 0, 1, 6, and 24 h following initial application of the isotopolog-labeled TRP and IFN-α2a. LC-MS metabolomics profiling was performed as described above and TRP catabolites were identified by their $^{13}C^{15}N$ peaks.

**Generation of shRNA and CRISPR cell lines.** To generate lentiviral particles, three million HEK293FT cells were transfected with 10 μg shIFNAR1 (TRCN0000059013, 5′-GCCAAGATTCAGGAAATTATT-3′), IFNAR1 gRNA (Human GeCKO Library A 22695, 5′-AACAGGAGCGATGAGTCTGT-3′), or IFNAR2 gRNA (Human GeCKO Library A 22697, 5′-GTGTATATCAGCCTCG TGTT-3′) plasmid, and 10 μg of packaging viral mix (3:1 pD8.9:pCMV-VSVG) using PEI in a 10 cm plate. Freshly collected lentiviral particles (2 mL) were then used to transduce 1.0e5 GM02767 cells in each well of a six-well plate. The cells were expanded to 10 cm plates before starting puromycin selection (1 μg/mL).

**IFN stimulations of fibroblasts and western blottings.** Cells were plated at equal densities (~33,000/cm$^2$) and were allowed to attach overnight. The next day, IFN-α2a (R&D Systems, 11101–2) was added at 10 ng/mL in RPMI (Corning, 10–040) and cells were collected by 0.25% trypsin (Gibco, 25200–056) at 0, 1, 6, and 24 h. Trypsin was quenched by adding RPMI with 10% Dialyzed FBS (Sigma, F0392). Cells were pelleted at $200 \times g$ for 5 min. Cell pellets were washed with 1× PBS, then resuspended in RIPA buffer containing 1 μg/mL pepstatin, 2 μg/mL aprotinin, 20 μg/mL trypsin inhibitor, 10 nM leupeptin, 200 nM Na3VO4, 500 nM phenylmethylsulfonyl fluoride, and 10 μM NaF. Suspensions were sonicated at 6 watts for 15 s two times and clarified by centrifugation at $21,000 \times g$ for 10 min at 4 °C. Supernatants were quantified in a Pierce BCA Protein Assay and diluted in complete RIPA with 4× Laemmli sample buffer. Tris-glycine SDS-polyacrylamide gel electrophoresis was used to separate 20–40 μg protein lysate, which was transferred to a 0.45 μm polyvinylidene fluoride membrane. Membranes were blocked in 5% non-fat dried milk or 2.5% bovine serum albumin (BSA) in Tris-buffered saline containing 0.1% TWEEN (TBS-T) at room temperature for 30 min. Immunoblotting was done using primary antibodies against IDO1 (1:1000, Cell Signaling Technology, Catalog Number 86630) and GAPDH (1:5000, Santa Cruz Biotechnology, Catalog Number 365062) overnight in 5% non-fat dried milk or 2.5% BSA in TBS-T at 4 °C, while shaking. Membranes were washed 3× in TBS-T for 5–15 min before probing with a horseradish peroxidase-conjugated secondary antibody in 5% non-fat dried milk at room temperature for 1 h. Membranes were again washed 3× in TBS-T for 5–15 min before applying enhanced chemiluminescence solution. Chemiluminensence signal was captured using a GE (Pittsburgh, PA) ImageQuant LAS4000.

**Quantitative reverse-transcription PCR.** Total RNA was collected from cells using TRIzol (Life Technologies/Thermo Fisher Scientific) and reverse transcription was carried out using the Applied Biosystems High Capacity cDNA kit (Life Technologies/Thermo Fisher Scientific). Quantitative PCR was carried out with reference to a standard curve using SYBR Select Master Mix for CFX (Life Technologies/Thermo Fisher Scientific) on a Viia7 Real-Time PCR system (Life Technologies/Thermo Fisher Scientific) and was normalized to 18S rRNA signals. Primers used for quantitative reverse-transcription PCR were as follows (5′ to 3′): 18S-F, GCCGCTAGAGGTGAAATTCTTG; 18S-R, CTTTCGCTCTGGTCCGTC TT; IDO1-F, CCGTAAGGTCTTGCCAAGAAATA; IDO1-R, GTCAGGGGCTT ATTAGGATCC; IFNAR1-F, GATTATCAAAAAACTGGGATGG; IFNAR1-R, C CAATCTGAGCTTTGCGAAATGG; IFNAR2-F, GGTTCTCATGGTGTATATC AGC; IFNAR2-R, GCAAGATTCATCTGTGTAATCAGG.

**Fibroblast TRP-tracing experiment data analysis.** Prior to plotting, tracing experiment data were batch-adjusted and measurements from biological replicates were summarized on time-course plots as the mean value of the replicates ± SEM. $P$-values were calculated in R with a Student's $t$-test and used the FDR method for multiple testing correction.

**Mesoscale Discovery assay.** The Mesoscale Discovery Human Biomarker 54-Plex was combined with U-Plex reagents for IFN-α2a, IFN-β, and IL-29 (IFN-λ), to measure a total of 55 unique cytokines from 500 μL of plasma from each of the 128 individuals in Cohort 3, as per the manufacturer's instructions. To visualize the effect of karyotype on each cytokine, the MSD data from both the D21 and T21 samples were z-scored using the mean and SD from the D21 group. Significantly differentially expressed cytokines were determined using the Kolmogorov–Smirnov test in R and an FDR-adjusted $p$-value threshold of 0.05. Prior to correlation analyses, both MSD data and metabolomics data were adjusted for age and sex using the linear model-fitting procedure described above. Spearman's rank correlations and $p$-values were calculated in R using the cor.test function. Adjusted $p$-values used FDR for multiple testing correction.

**Mouse strains and data analysis.** Dp(10Prmt2-Pdxk)1Yey/J (Dp(10)1Yey/ + ), Dp(16Lipi-Zbtb21)1Yey/J (Dp(16)1Yey/ + ), and Dp(17Abcg1-Rrp1b)1(Yey)/J (Dp(17)1Yey/ + ) have been previously described[46]. Dp(16) mice were purchased from Jackson Laboratories or provided by Drs Faycal Guedj and Diana Bianchi at the National Institutes of Health. Dp(10)1Yey/ + , and Dp(17)1Yey/ + mice were provided by Drs Katheleen Gardiner and Santos Franco, respectively. Animals were used between 6 and 30 weeks of age. All mice were maintained on a C57Bl/6 background and housed in specific pathogen-free conditions. All experiments were approved by the Institutional Animal Care and Use Committee at the University of Colorado Anschutz Medical Campus. Prior to plotting and testing for statistical significance, mouse metabolomics data were adjusted for batch and sex using the linear modeling procedure described above. $P$-values were calculated in R with a Student's $t$-test and used FDR for multiple testing correction.

**Reporting summary.** Further information on research design is available in the Nature Research Reporting Summary linked to this article.

## Data availability

The source data for Fig. 3a, b, f, g are provided as a Source Data file. All other figures were generated using data proved in Supplementary Data and/or deposited into publicly available databases. RNA-seq data were deposited at the Gene Expression Omnibus (National Center for Biotechnology Information) with series accession number GSE128622. Metabolomics data were deposited in the Metabolomics Workbench database with Study IDs ST001240, ST001241, ST001242, and ST001243. The data and code at https://github.com/CostelloLab/Trisomy21_KYN_metabolomics can be used to generate Figs. 1, 2, and 4, as well as the metabolomics portions of Fig. 3.

## Code availability

All data and R code developed for the full set of statistical metabolomic analyses is available at: https://github.com/CostelloLab/Trisomy21_KYN_metabolomics. To ensure reproducibility, this code can be run to generate all figures for this manuscript.

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

## Acknowledgements

We thank the University of Colorado Cancer Center Human Immune Monitoring Shared Resource for assistance with Mesoscale Discovery Assay studies. We thank the individuals with Down syndrome that donated the biological samples, which enabled these studies. This work was supported by NIH grants R01AI150305, R01AI145988, T15LM009451, T32CA190216, UL1TR002535, and R01HL137990-02S1, the Linda Crnic Institute for Down Syndrome, the Global Down Syndrome Foundation, the Anna and John J. Sie Foundation, the Boettcher Foundation, and the University of Colorado School of Medicine. The Functional Genomics Facility and Metabolomics Core are shared resources of the University of Colorado Cancer Center supported by NIH grant P30 CA046934. This work was also supported by research grants from the Carlos III Institute of Health, Spain (grants PI14/01126 and PI17/01019 to J.F., PI13/01532 and PI16/01825 to R.B., PI14/1561 and PI17/01896 to A.L., PI18/00335 to M.C.-I.), the National Institutes of Health (NIA grants 1R01AG056850–01A1; R21AG056974 and R01AG061566) and the CIBERNED program (Program 1, Alzheimer Disease to A.L. and SIGNAL study, www.signalstudy.es), partly funded by Fondo Europeo de Desarrollo Regional (FEDER), Unión Europea, "Una manera de hacer Europa." This work has also been supported by a "Marató TV3" grant (20141210 to J.F. and 044412 to R.B.) and by Generalitat de Catalunya (SLT006/17/00119 to J.F.), and a grant from the Fundació Bancaria La Caixa to R.B.

## Author contributions

R.K.P., J.C.C., K.D.S., and J.M.E., conception and design, acquisition of data, analysis and interpretation of data, drafting or revising the article. A.L., R.B., J.F., and A.D., conception and design, acquisition of data, analysis and interpretation of data. R.C.H., M.P.L., K.P.S., K.A.W., R.M., K.D.T., H.C.L., A.L.R., R.E.G., M.C., R.B.W., D.E.K., M.J., acquisition of data, analysis, and interpretation of data.

## Competing interests

The authors declare no competing interests.

## Additional information

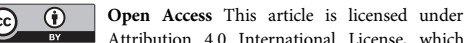

