## [Peer Review File · Nature Communications]

Reviewers' comments:

Reviewer #1, expert on kynurenine pathway in immune response (Remarks to the Author):

In this work, Powers et al investigate the potential role of the kynurenine pathway (KP) in T21-associated pathology. Interestingly, the authors detect increased KP activation in T21, which they attribute to the upregulation of IDO1 as a result of increased IFN-I signaling resulting from IFNAR duplication. These are interesting, well performed studies. However, it is my opinion that some points should be clarified before this work is ready for publication.

1. Did the authors detect changes in the expression of IL-27, which is induced by IFN-I and is a known driver of IL-10 expression?
2. What are the levels of IFN-I/IFN-II/ IFN-III detected in D21 and T21 patients? Those should be shown in the main manuscript.
3. How do the increased levels of pro-inflammatory cytokines, IFNs and IL-10 compare with those detected in infectious and autoimmune diseases?
4. Figs. 3c-e show each 2 panels, one for D21 cells and one for T21 cells. Since the central point of the manuscript is a potential upregulation of KP in T21, data obtained in D21 and T21 cells should be shown in the same panel, and a direct statistical comparison between D21 and T21 cells (resting or IFN-treated) should be performed to demonstrate statistical significant differences in Trp, Kyn and Trp/Trp levels.
5. In the experiments shown in Figs. 3f,g, what is the source of IFNs leading to the IFNAR-driven increase in IDO expression in T21 cells?
6. Do the authors detect changes in the levels of KP metabolites in CSF T21 samples or in the CNS of Dp16 mice? This is important to postulate that KP metabolites have any effect on the CNS.
7. Although the authors shortly discuss this at the end, it is challenging to understand how the increased levels of anti-inflammatory Kyn are linked to the increased autoimmune and neuroinflammatory conditions linked to T21. This should be better explained in the discussion, and alternative interpretations should be provided.

Reviewer #2, expert in Down syndrome (Remarks to the Author):

People with DS are affected by numerous pathologies including cognitive impairment and predispositions to Alzheimer's disease and blood cancer. Recent studies by Espinosa and colleagues have shown that trisomy 21 is associated with constitutive activation of the interferon response and chronic inflammation. Here they extend their recent transcriptomics and plasma proteomics studies of cohorts of individuals with or without DS to metabolomics. Among the altered metabolic pathways, their study revealed activation of the kynurenine pathway, with high levels of kynurenine and quinolinic acid. They also observed increased expression of the tryptophan catabolic enzyme IDO1 in individuals with trisomy 21 by RNA-sequencing. Given that IDO1 is an IFN stimulated gene, the authors propose a pathway by which increased expression of the IFN receptors encoded on chromosome 21 leads to increased IDO1 expression and trp metabolism. This model is supported by the observation that a mouse model of DS with the four IFN receptors, but not two strains with smaller regions of trisomy, show a similar enrichment of kynurenine.

This is an interesting and important study, which furthers our knowledge of the link between trisomy

21 and the pathologies of DS. The data are clearly presented and described.

1. The data in figure 4 supports the argument that expression of the four IFN receptors has a downstream effect on kynurenine levels. Is IDO1 expression increased in these mice? What accounts for the high degree of variation in kynurenine levels within the Dp16 strain? What are the differences in the neurologic phenotypes between Dp16 and the other two strains with smaller trisomy?

2. I think the discussion is a bit too speculative. How likely is it that this pathway explains so many of the pathologies of DS?

3. Have IDO1 inhibitors been given to any of the mouse models of DS?

Minor comments:

1. The authors could include the names of the other metabolites that are highly increased or decreased in Figure 1A.

2. It would be helpful if the authors could note the location of the four IFN receptors in the Dp16 model. Are there any genes in the Dp17 or Dp10 mice that have been implicated in Trp catabolism?

3. In figure 3f, it looks as though the order of target genes is altered in the T21-IFNAR1/2 KO group. (ie the second bar is white instead of gray).

Reviewer #3, expert in metabolomics (Remarks to the Author):

The manuscript by Powers et al, investigating the plasma metabolome associated with Trisomie 21 is an interesting and well written manuscript.

I have several major and minor concerns which would need to be addressed before publication can be advised.

Major:

Why was such a high metabolite cut off >90% applied. As the authors have two groups this applies that only metabolites found in Ctrl as well as Disease will be detected, but are you not limiting your view to find a particularly disease associated metabolite in this way? In this case the metabolite would only be present in 50% of all samples...

In the method section the authors specifically point out the extraction of oxylipids? Why? Can they be even detected by an untargeted metabolomics approach? Later in the manuscript these metabolites do not play any role anymore, where is this link coming from?

I have to say that I find the Metabolomics method section rather confusing. The authors state: Compounds of interest were monitored and quantified in negative ion mode against deuterium-labeled internal standards. Then I am referred to two papers with a totally different gradient etc. and these internal standards are not given?! So, all the 91 metabolites were quantified against internal standards? Frankly, I am a bit lost, what was now exactly done, the gradient extraction etc is clear to me, but which quantification was used? Absolutes quant of all metabolites? Which internal standards for which metabolites? When I look at the figures its adjusted intensity, that doesn't look like real concentrations to me? Sorry but I really find this confusing. Here it would also be important that the authors wold carry our absolute quantification of Kyn in there samples. Such data will be very valuable to other researchers and gives important insights about concentrations to be used in biological experiments. I would strongly advice to abolutely quantify at least Kyn.

For transcriptome analysis a third cohort was used, how can this be justified, does this influence data comparability?

The "flux" (its not a real flux experiment which would allow determination of kinetics) analysis with labeled Trp is not well described, which isotopes were used, how were the metabolites exactly detected...?

The authors focused entirely on Trp metabolism by IDO. However, when Trp is metabolized to Kyn by IDO an important feature has not been investigated, namely the production of formic acid. In other words, while obviously much is known about Kyn and Trp, a third yet not investigated player, formic acid might also play a role in the puzzle. How can effects of formic acid be ruled out? Maybe changes in pH or formic acid itself play a role in the immune system? Especially in the light of KP I wonder if the production of formic acid in the nervous system might not by its own present a pathophysiological mechanism? This discussion evolving around formic acid was unfortunately entirely circumvented in the discussion section. Along these lines I wonder if also increased formic acid levels were detected in the authors data set?

Minor:

Why is the organic solvent mixture for extraction called lysis buffer? There is no buffer to be found in this mixture in my view.

Please used the times sign where appropriate and do not substitute with the letter x

Consistency with units, L or l?

Don't forget the degrees sign for degrees C

Response to Reviewers' Comments

We are very grateful for the Reviewers' constructive comments, which prompted many new experiments and analyses, as well as significant revisions of the text. We hope the Reviewers' will find the revised manuscript to be much improved and meritorious of publication.

Reviewer #1 (expert on kynurenine pathway in immune response):

In this work, Powers et al investigate the potential role of the kynurenine pathway (KP) in T21-associated pathology. Interestingly, the authors detect increased KP activation in T21, which they attribute to the upregulation of IDO1 as a result of increased IFN-I signaling resulting from IFNAR duplication. These are interesting, well performed studies. However, it is my opinion that some points should be clarified before this work is ready for publication.

Comment #1: *Did the authors detect changes in the expression of IL-27, which is induced by IFN-I and is a known driver of IL-10 expression?*

Response: We thank the reviewer for this comment, which prompted us to generate more data to investigate the interplay between IL-10, IL-27, IFN signaling and KP dysregulation, leading to new interesting observations. Of note, this revised manuscript describes MSD profiling of 55 cytokines in a cohort of 128 research participants, 74 of which have T21, thus **doubling the number of samples analyzed for cytokine levels** relative to the original submission. Importantly, the same plasma samples were used for **absolute quantification** of key metabolites in the KP as suggested by **Reviewer #3**, which enabled us to extend our correlation analysis between KP dysregulation and cytokine levels. Interestingly, in this extended analysis, IL-10 is the most significantly elevated cytokine in plasma of people with Down syndrome (followed by IP-10, IL-6, IL-22 and TNF- α , **Fig. 2b** and **Supplementary Data 8**); and the second most positively correlated with the KYN/TRP ratio after IP-10 (**Fig. 2c** and **Supplementary Data 10**). However, in this same analysis, IL-27 levels were not significantly elevated in people with DS (**Supplementary Data File 8**), indicating an alternative mechanism of IL-10 upregulation in Down syndrome. In fact, when we performed a correlation analysis between IL-10 and all other cytokines measured, the top cytokines positively correlated with IL-10 were TNF- α , IP-10 and MCP-1, with IL-27 not being significantly correlated (shown here in **Figure R1** for Reviewer's only). Therefore, while it seems that IL-10 is positively correlated with IFN signaling and dysregulation of the KP, the correlation with IL-27 is less obvious in our study. The revised manuscript provides a more thorough description of the correlations between cytokines and KP dysregulation.

Figure R1. IL-10 levels correlate positively with TNF- α , IP-10 and MCP-1, but not so with IL-27. Spearman's rank correlation analyses were performed using MSD data for the indicated cytokines in a cohort of 128 research participants, 74 of them with trisomy 21 (T21) and 54 typical controls (D21).

Comment #2: *What are the levels of IFN-I/IFN-II/IFN-III detected in D21 and T21 patients? Those should be shown in the main manuscript.*

Response: Our extended MSD cytokine profiling included measurement of four IFN ligands, IFN- α 2a and IFN- β (IFN-I), IFN- γ (IFN-II), and IL-29 (IFN- λ 1, IFN-III). Levels of IFN- α 2a, IFN- β , and IFN- γ were below the limit of detection for many of the samples, which is not uncommon for these labile, low abundance cytokines. Nevertheless, IFN- α 2a did pass our statistical cutoff as elevated in people with Down syndrome, but IFN- β , IFN- γ , IFN- λ 1 did not. All four cytokines are now discussed in the body of the manuscript and data shown in

Supplementary Fig. 3. As noted in the original manuscript, IFN- λ 1 correlated positively with KP dysregulation. Thanks to our expanded cohort used for cytokine profiling and absolute quantification of KP metabolites explained above, we are now able to demonstrate that levels of IFN- γ , IFN- α 2a, and IFN- λ 1 correlate positively with the KYN/TRP ratio (**Fig. 2d**). In the revised manuscript, we further emphasize the concept that hyperactive IFN signaling in Down syndrome is likely to be driven by the extra copy of the IFN receptors, which in turn could lead to elevated levels of the ligands themselves in a 'fast forward loop' as proposed by Kirsammer and Crispino¹. In fact, the profound impact of the extra IFNR copy is demonstrated in the cell-based assays shown in **Fig. 3**.

Comment #3: *How do the increased levels of pro-inflammatory cytokines, IFNs and IL-10 compare with those detected in infectious and autoimmune diseases?*

Response: We appreciate this comment by the reviewer which aligns precisely with our ongoing research interests. In order to answer this question, we needed to compare our analysis of plasma from people with DS with studies utilizing the same blood processing protocol and MSD platform that we have used. An extensive literature search identified a study of plasma from people with systemic lupus erythematosus (SLE), an IFN-driven autoimmune disorder, where the authors measured 11 of the of the same cytokines that we measured, using the same platform². A comparison of our findings with data from that study revealed that all 11 cytokines commonly measured in both studies are significantly elevated in both conditions: IL-10, IP-10, IL-6, TNF- α , MCP-1, CRP, VEGF, IL-15, IL-8, Eotaxin, and TARC (**Supplementary Data 9**). Direct comparison of absolute levels of these proteins often revealed markedly lower levels in our control population relative to the control population of the SLE study. This could be due to the fact that the control population from the SLE study was substantially older than ours with a median age of 48.2 compared to 27.5. The SLE control population was also 92% female, which is another potentially confounding difference. Nonetheless, for three proteins with similar levels in the control population, we found that levels in the T21 population were not as elevated as for the SLE population. For example, IL-10 levels were ~0.3 pg/mL for both control populations, but ~0.6 pg/mL for T21 as opposed to ~0.8 pg/mL for SLE. For MCP-1, control levels were ~60 pg/mL in our study and ~69 pg/mL in the SLE study, rising to ~76 pg/mL and ~110 pg/mL, respectively. Finally, for IL-15, control levels in the two studies were ~2.1 pg/mL, and elevated to ~2.5 pg/mL for T21 and ~2.9 pg/mL for SLE. A description of these results is now included in the revised manuscript and all data shown in **Supplementary Data 9**.

Encouraged by this comment, and also in response to **Reviewer #3**, we performed a similar comparison for absolute levels of KYN and QA in plasma and CSF, which revealed that levels of KYN in DS are higher than those found in Alzheimer's disease, SLE and depression, and on par with levels seen during lethal cardiac arrest (see below response to **Reviewer #3** below).

Comment #4: *Figs. 3c-e show each 2 panels, one for D21 cells and one for T21 cells. Since the central point of the manuscript is a potential upregulation of KP in T21, data obtained in D21 and T21 cells should be shown in the same panel, and a direct statistical comparison between D21 and T21 cells (resting or IFN-treated) should be performed to demonstrate statistical significant differences in Trp, Kyn and Trp/Trp levels.*

Response: We thank the Reviewer for this comment, and we have modified the figures accordingly for greater clarity.

Comment #5: *In the experiments shown in Figs. 3f,g, what is the source of IFNs leading to the IFNAR-driven increase in IDO expression in T21 cells?*

Response: In the referenced figures, recombinant human IFN- α 2a is exogenously provided. We have amended the text to make this more explicit.

Comment #6: *Do the authors detect changes in the levels of KP metabolites in CSF T21 samples or in the CNS of Dp16 mice? This is important to postulate that KP metabolites have any effect on the CNS.*

Response: Encouraged by this comment, we set out to address this very difficult question by seeking out potential collaborators with hard-to-obtain CSF samples from people with Down syndrome and typical controls.

We are happy to report that Dr. Juan Fortea and his team at the St. Pau Hospital in Barcelona, Spain, kindly agreed to provide us with 100 CSF samples, 50 of them from individuals with T21, which we analyzed using the absolute quantification method requested by **Reviewer #3** (see below). Reassuringly, these efforts revealed significantly elevated levels of KYN, L-formyl-KYN, QA, and the KYN/TRP ratio in the CSF of people with Down syndrome (new **Fig. 1f** and **Supplementary Data 6**). This result highlights the potentially deleterious impact of KYN dysregulation in the CNS of those with T21. These findings agree with a recently published paper showing that plasma levels of KYN correlate strongly with levels in the CSF³. Naturally, this important contribution by the Fortea team is now acknowledged by inclusion of additional authors. Regarding the comment about KYN levels in animal models of DS, the Patterson group at the University of Denver recently published a report demonstrating elevated KYN levels in the brain tissue of Ts65Dn mouse model of Down syndrome and that these levels can be lowered by the anti-inflammatory drug rapamycin⁴. All of these results are now discussed in the revised manuscript.

Comment #7: *Although the authors shortly discuss this at the end, it is challenging to understand how the increased levels of anti-inflammatory Kyn are linked to the increased autoimmune and neuroinflammatory conditions linked to T21. This should be better explained in the discussion, and alternative interpretations should be provided.*

Response: We have updated the Discussion following guidance from both Reviewer #1 and Reviewer #2. KYN has complex effects on immune cells, leading to suppression of effector CD4 and CD8 T cells, while also inducing regulatory T cells (Tregs, reviewed in⁵) and it may represent an immunosuppressive negative feedback mechanism downstream of exacerbated IFN signaling. In fact, Tregs have been shown to be much elevated in the circulation of people with DS⁶, which could contribute to both autoimmune conditions and poor resolution of inflammatory processes. Given that people with DS are highly predisposed to myriad autoimmune disorders⁷⁻¹¹ and increased mortality due to *pneumonia* lung infections¹², our results justify future investigations to define the role of KP dysregulation (versus alternative mechanisms) in the alteration of diverse immune cell types in particular, and the development of common co-morbidities in DS in general.

Reviewer #2 (expert in Down syndrome):

People with DS are affected by numerous pathologies including cognitive impairment and predispositions to Alzheimer's disease and blood cancer. Recent studies by Espinosa and colleagues have shown that trisomy 21 is associated with constitutive activation of the interferon response and chronic inflammation. Here they extend their recent transcriptomics and plasma proteomics studies of cohorts of individuals with or without DS to metabolomics. Among the altered metabolic pathways, their study revealed activation of the kynurenine pathway, with high levels of kynurenine and quinolinic acid. They also observed increased expression of the tryptophan catabolic enzyme IDO1 in individuals with trisomy 21 by RNA-sequencing. Given that IDO1 is an IFN stimulated gene, the authors propose a pathway by which increased expression of the IFN receptors encoded on chromosome 21 leads to increased IDO1 expression and trp metabolism. This model is supported by the observation that a mouse model of DS with the four IFN receptors, but not two strains with smaller regions of trisomy, show a similar enrichment of kynurenine.

This is an interesting and important study, which furthers our knowledge of the link between trisomy 21 and the pathologies of DS. The data are clearly presented and described.

Comment #1: *The data in figure 4 supports the argument that expression of the four IFN receptors has a downstream effect on kynurenine levels. Is IDO1 expression increased in these mice? What accounts for the high degree of variation in kynurenine levels within the Dp16 strain? What are the differences in the neurologic phenotypes between Dp16 and the other two strains with smaller trisomy?*

Response: Encouraged by this comment, we decided to monitor *Ido1* expression in the mouse models. Given the short time frame for revisions relative to the timeline required for new animal experiments, we restricted this analysis to wild type and Dp16 mice, both under 'normal' conditions and after acute exposure to poly(I:C), a dsRNA and TLR agonist known to induce IFN signaling. We then measure *Ido1* protein expression in the spleen of these mice. This exercise demonstrated variable yet increased *Ido1* expression in the Dp16 model,

which is further elevated by poly(I:C) injection. However, the difference did not reach statistical significance at the current sample size, which could be explained in part by the fact that ‘sham’ treatment involved intraperitoneal injections, which in themselves could trigger an inflammatory response. We show these results here in **Fig. R2** for Reviewer’s only. Given that the Patterson group recently demonstrated elevated levels of KYN in brain tissue of the Ts65Dn model of Down syndrome⁴, we decided not to include these results in the manuscript, choosing instead to decipher the role of *Ido1* expression and KP dysregulation in mouse models in a follow up study. Nevertheless, following Reviewer’s guidance, we explicitly mention the high variability observed, as well as the results in the Patterson et al paper in the revised manuscript. We have also included a brief description of the neurological phenotypes that have been directly compared among the Dp10, Dp16, and Dp17 mouse strains, revealing that Dp16 mice have demonstrated defects in spatial learning and memory, context-associated learning, and decreased hippocampal long-term potentiation, not found in Dp10 or Dp17 mice¹³. Of note, QA is a potent neurotoxin that acts as an excitotoxic *agonist* of NMDA receptors, which could explain the beneficial effects of memantine, an NMDA *antagonist* that protects from QA-mediated neurotoxicity, in animal models of DS carrying triplication of the IFNRs¹⁴⁻¹⁶.

Figure R2. Measurements of Ido1 protein expression in the spleen of wild type and Dp16 mice injected with vehicle (sham) or with poly(I:C) at 10 µg/g of body weight. Spleens were harvested 24 hours after injection. N=3 per group. Although Ido1 expression is elevated in the Dp16 mice and after pI:C treatment, the differences do not reach statistical significance at this sample size.

Comment #2: *I think the discussion is a bit too speculative. How likely is it that this pathway explains so many of the pathologies of DS?*

Response: We have updated the discussion following guidance from both **Reviewer #1** and **Reviewer #2** to focus on discussion of those comorbidities with the strongest link to KP dysregulation. We also acknowledge in the revised discussion that a causal role for KP dysregulation in the higher prevalence of these comorbidities in people with DS will require extensive future research efforts relying on both human samples and mouse models. Our team is working arduously to secure the necessary funding for these efforts.

Comment #3: *Have IDO1 inhibitors been given to any of the mouse models of DS?*

Response: This is a very exciting experiment that has not, to the best of our knowledge, been done in mouse models of Down syndrome, but that is part of our future plans. We are encouraged by the fact that IDO1 inhibitors have been shown to improve cognitive deficits in mouse models of AD¹⁷ and neuroinflammation-induced depression¹⁸. Although several IDO1 inhibitors have been developed, we believe that Epacadostat, currently being developed for cancer therapy, would be most suitable for these studies. However, it should be noted that IDO1 inhibition in people with Down syndrome may disrupt the negative feedback loop involving IDO1 and KYN to resolve inflammation. Therefore, we believe a more suitable strategy to counteract the deleterious impact of the IFN>IDO1>KYN axis will be to inhibit this pathway more broadly at an upstream step using JAK inhibitors. In fact, we recently published the first two known cases of JAK inhibition for therapeutic purposes in people with Down syndrome affected by the autoimmune hair loss condition, alopecia areata¹⁹. In fact, our team is currently involved in the testing of JAK inhibitors in mouse models of Down syndrome and in the launching of a clinical trial for JAK inhibition in Down syndrome, which will include measurements of potential cognitive improvement. We look forward to completing these efforts in the near future and publish the results of these investigations.

Comment #4: *Minor comment: The authors could include the names of the other metabolites that are highly increased or decreased in Figure 1A.*

Response: We have amended the figure to include labels for many additional metabolites.

Comment #5: *Minor comment: It would be helpful if the authors could note the location of the four IFN receptors in the Dp16 model. Are there any genes in the Dp17 or Dp10 mice that have been implicated in Trp catabolism?*

Response: We have added the IFN receptor gene cluster to the schematic in **Fig. 4**. Our detailed analysis of the genes triplicated in these mouse model failed to identify additional genes associated with the KYN pathway.

Comment #6: *Minor comment: In figure 3f, it looks as though the order of target genes is altered in the T21-IFNAR1/2 KO group. (ie the second bar is white instead of gray).*

Response: We thank the Reviewer for noting this error and we have corrected this figure.

Reviewer #3 (expert in metabolomics):

The manuscript by Powers et al, investigating the plasma metabolome associated with Trisomy 21 is an interesting and well written manuscript. I have several major and minor concerns which would need to be addressed before publication can be advised.

Comment #1: *Why was such a high metabolite cut off >90% applied. As the authors have two groups this applies that only metabolites found in Ctrl as well as Disease will be detected, but are you not limiting your view to find a particularly disease associated metabolite in this way? In this case the metabolite would only be present in 50% of all samples...*

Response: We thank the reviewer for this comment, which prompted us to improve the description of our methodology for filtering metabolites based on their presence in groups of different karyotypes but also across different cohorts. When analyzing Cohorts 1 and 2 *individually*, we retained only metabolites that were detected in >90% of **either** karyotype. This cut-off choice could eventually enable us to study metabolites more detectable in *either* the control or disease group. Then, when analyzing Cohorts 1 and 2 in *combination*, we retained metabolites that were detected in **both** cohorts by the criteria described above, leading to our final list of 91 KEGG-annotated metabolites shown in **Fig. 1a**. This way, we could focus our study on differential metabolites present in more than one cohort. This is now better explained in the revised text.

Comment #2: *In the methods section the authors specifically point out the extraction of oxylipids? Why? Can they be even detected by an untargeted metabolomics approach? Later in the manuscript these metabolites do not play any role anymore, where is this link coming from?*

Response: We have clarified the methods section to remove mention of oxylipids.

Comment #3: *I have to say that I find the Metabolomics method section rather confusing. The authors state: Compounds of interest were monitored and quantified in negative ion mode against deuterium-labeled internal standards. Then I am referred to two papers with a totally different gradient etc. and these internal standards are not given?! So, all the 91 metabolites were quantified against internal standards? Frankly, I am a bit lost, what was now exactly done, the gradient extraction etc is clear to me, but which quantification was used? Absolutes quant of all metabolites? Which internal standards for which metabolites? When I look at the figures its adjusted intensity, that doesn't look like real concentrations to me? Sorry but I really find this confusing. Here it would also be important that the authors would carry out absolute quantification of Kyn in their samples. Such data will be very valuable to other researchers and gives important insights about concentrations to be used in biological experiments. I would strongly advice to absolutely quantify at least Kyn.*

Response: We thank the reviewer for this comment regarding the explanation of our methodology and apologize for the confusion. We have now clarified explicitly that the quantification of metabolites from Cohorts 1 and 2 that are shown in **Fig. 1** and **Supplementary Fig. 1** are *relative quantifications* that have not been

quantified against internal standards. We have also expanded the methods section to more fully describe all metabolomics experiments performed in this manuscript. Furthermore, in response to the Reviewer's comment about absolute quantification, our team embarked on significant efforts to create a **third, much larger cohort of plasma samples** which were **subjected to absolute quantification using $^{13}\text{C}_{10}$ KYN and $^{15}\text{N}_2$ TRP as standards**. Analysis of this cohort of 128 research participants (74 with trisomy 21) not only confirmed dysregulation of the KP, but also enabled us to compare levels of key TRP metabolites to those observed in studies of SLE, AD, depression, and heart attack, that used similar methodology. This exercise revealed that KYN levels in our **control group** are lower than those in other studies, whereas the levels detected for people with Down syndrome are **equal or higher** than each of these disease groups. These data are now included in expanded **Fig. 1** and **Supplementary Data 4**. A thorough description of absolute quantification methods and calculations has also been added. Finally, we applied this absolute quantification method to the analysis of CSF samples as requested by **Reviewer #1**, which once again confirmed KP dysregulation in the CNS of people of with DS.

Comment #4: *For transcriptome analysis a third cohort was used, how can this be justified, does this influence data comparability?*

Response: Our cohort selection for different -omics analyses is often driven by experimental limitations, as it is not always possible to obtain enough white blood cells for flow cytometry, RNA-seq analysis and other studies requiring white blood cells. With that being said, our newly introduced third plasma metabolomics cohort (Cohort 3) included nine individuals for which the transcriptome analysis was previously performed, facilitating a direct correlation of IDO1 levels with KP activation (**Fig. R3** shown here for Reviewers only). Indeed, there is a positive correlation between IDO1 levels, albeit not statistically significant with this small sample size. Nevertheless, our expanded cytokine analysis on matched samples for which we performed the absolute quantification of KYN metabolites shows that IP-10 (Interferon-inducible protein 10), is the top cytokine correlated with elevated KYN/TRP ratio (new **Fig. 2d** and **Supplementary Data 10**), thus reinforcing the notion that dysregulation of the KP in Down syndrome is associated with increased IFN signaling.

Figure R3. Correlation between KYN levels in plasma and IDO1 mRNA expression in white blood cells of the same individuals.

Comment #5: *The "flux" (its not a real flux experiment which would allow determination of kinetics) analysis with labeled Trp is not well described, which isotopes were used, how were the metabolites exactly detected...?*

Response: We agree that this experiment could have been better described. We have now expanded description of this experiment and clarified that it is a tracing experiment rather than 'flux'.

Comment #6: *The authors focused entirely on Trp metabolism by IDO. However, when Trp is metabolized to Kyn by IDO an important feature has not been investigated, namely the production of formic acid. In other words, while obviously much is known about Kyn and Trp, a third yet not investigated player, formic acid might also play a role in the puzzle. How can effects of formic acid be ruled out? Maybe changes in pH or formic acid itself play a role in the immune system? Especially in the light of KP I wonder if the production of formic acid in the nervous system might not by its own present a pathophysiological mechanism? This discussion evolving around formic acid was unfortunately entirely circumvented in the discussion section. Along these lines I wonder if also increased formic acid levels were detected in the authors data set?*

Response: We thank the Reviewer for these insightful comments. Unfortunately, formic acid is not detectable with our current methodology, as our sample processing protocol involves addition of formic acid itself. However, our literature search revealed that Caracausi et al²⁰, found elevated levels of formate in the plasma of people with Down syndrome, which they attributed to alterations in the TCA cycle. Thanks to the Reviewer's

comment, we now mention this interesting result and its potential relationship to dysregulation of the KYN pathway in the Discussion.

Comment #7: Minor comment: Why is the organic solvent mixture for extraction called lysis buffer? There is no buffer to be found in this mixture in my view.

Response: Thanks for this comment. We have corrected the description of the method.

Comment #8: Please used the times sign where appropriate and do not substitute with the letter x. Consistency with units, L or l? Don't forget the degrees sign for degrees C.

Response: Thank you. We have edited these for consistency throughout.

References.

- 1 Kirsammer, G. & Crispino, J. D. Signaling a link between interferon and the traits of Down syndrome. *Elife* **5**, doi:10.7554/eLife.20196 (2016).
- 2 Idborg, H. *et al.* TNF-alpha and plasma albumin as biomarkers of disease activity in systemic lupus erythematosus. *Lupus Sci Med* **5**, e000260, doi:10.1136/lupus-2018-000260 (2018).
- 3 Jacobs, K. R. *et al.* Correlation between plasma and CSF concentrations of kynurenine pathway metabolites in Alzheimer's disease and relationship to amyloid-beta and tau. *Neurobiology of aging* **80**, 11-20, doi:10.1016/j.neurobiolaging.2019.03.015 (2019).
- 4 Duval, N., Vacano, G. N. & Patterson, D. Rapamycin Treatment Ameliorates Age-Related Accumulation of Toxic Metabolic Intermediates in Brains of the Ts65Dn Mouse Model of Down Syndrome and Aging. *Front Aging Neurosci* **10**, 263, doi:10.3389/fnagi.2018.00263 (2018).
- 5 Routy, J. P., Routy, B., Graziani, G. M. & Mehraj, V. The Kynurenine Pathway Is a Double-Edged Sword in Immune-Privileged Sites and in Cancer: Implications for Immunotherapy. *Int J Tryptophan Res* **9**, 67-77, doi:10.4137/IJTR.S38355 (2016).
- 6 Pellegrini, F. P. *et al.* Down syndrome, autoimmunity and T regulatory cells. *Clin Exp Immunol* **169**, 238-243, doi:10.1111/j.1365-2249.2012.04610.x (2012).
- 7 Soderbergh, A. *et al.* Autoantibodies linked to autoimmune polyendocrine syndrome type I are prevalent in Down syndrome. *Acta paediatrica (Oslo, Norway : 1992)* **95**, 1657-1660, doi:10.1080/08035250600771466 (2006).
- 8 Chen, M. H., Chen, S. J., Su, L. Y. & Yang, W. Thyroid dysfunction in patients with Down syndrome. *Acta Paediatr Taiwan* **48**, 191-195 (2007).
- 9 Sureshbabu, R. *et al.* Phenotypic and dermatological manifestations in Down Syndrome. *Dermatology online journal* **17**, 3 (2011).
- 10 Marild, K. *et al.* Down syndrome is associated with elevated risk of celiac disease: a nationwide case-control study. *J Pediatr* **163**, 237-242, doi:10.1016/j.jpeds.2012.12.087 (2013).
- 11 Alexander, M. *et al.* Morbidity and medication in a large population of individuals with Down syndrome compared to the general population. *Developmental medicine and child neurology* **58**, 246-254, doi:10.1111/dmcn.12868 (2016).
- 12 Englund, A., Jonsson, B., Zander, C. S., Gustafsson, J. & Anneren, G. Changes in mortality and causes of death in the Swedish Down syndrome population. *American journal of medical genetics. Part A* **161A**, 642-649, doi:10.1002/ajmg.a.35706 (2013).
- 13 Yu, T. *et al.* Effects of individual segmental trisomies of human chromosome 21 syntenic regions on hippocampal long-term potentiation and cognitive behaviors in mice. *Brain research* **1366**, 162-171, doi:10.1016/j.brainres.2010.09.107 (2010).
- 14 Costa, A. C., Scott-McKean, J. J. & Stasko, M. R. Acute injections of the NMDA receptor antagonist memantine rescue performance deficits of the Ts65Dn mouse model of Down syndrome on a fear conditioning test. *Neuropsychopharmacology* **33**, 1624-1632, doi:10.1038/sj.npp.1301535 (2008).
- 15 Lockrow, J., Boger, H., Bimonte-Nelson, H. & Granholm, A. C. Effects of long-term memantine on memory and neuropathology in Ts65Dn mice, a model for Down syndrome. *Behav Brain Res* **221**, 610-622, doi:10.1016/j.bbr.2010.03.036 (2011).

- 16 Scott-McKean, J. J. & Costa, A. C. Exaggerated NMDA mediated LTD in a mouse model of Down syndrome and pharmacological rescuing by memantine. *Learn Mem* **18**, 774-778, doi:10.1101/lm.024182.111 (2011).
- 17 Yu, D. *et al.* The IDO inhibitor coptisine ameliorates cognitive impairment in a mouse model of Alzheimer's disease. *J Alzheimers Dis* **43**, 291-302, doi:10.3233/JAD-140414 (2015).
- 18 Dobos, N. *et al.* The role of indoleamine 2,3-dioxygenase in a mouse model of neuroinflammation-induced depression. *J Alzheimers Dis* **28**, 905-915, doi:10.3233/JAD-2011-111097 (2012).
- 19 Rachubinski, A. L. *et al.* Janus kinase inhibition in Down syndrome: 2 cases of therapeutic benefit for alopecia areata. *JAAD Case Rep* **5**, 365-367, doi:10.1016/j.jdc.2019.02.007 (2019).
- 20 Caracausi, M. *et al.* Plasma and urinary metabolomic profiles of Down syndrome correlate with alteration of mitochondrial metabolism. *Scientific reports* **8**, 2977, doi:10.1038/s41598-018-20834-y (2018).

Reviewers' comments:

Reviewer #1 (Remarks to the Author):

The authors have addressed my comments. One minor point is that the authors may want to refer in the discussion to previous work linking IFN, Kyn, AHR and IL-10. Rothhammer et al (Nat medicine 2016) showed that AHR expression is induced by IFN-I, and AHR mediates many of the physiological effects of kynurenine including IL-10 production and the control of peripheral and CNS inflammation (Rothhammer et al 2019, NRI).

Reviewer #2 (Remarks to the Author):

The authors have addressed my concerns.

Reviewer #3 (Remarks to the Author):

The authors have improved their experimental section. However, and yes, this might be me, but I still dont think the described metabolomics experiments can be repeated with the given details.

The authors indicate to have run their samples on 3 platforms and then? You combined the data? You ran data analysis separately and then ended up with the 91 metabolites significantly changed? You did rel. quant on all 3 data sets? So which is the data set I am looking at in your figures? Were different matrices run with different methods? How comparable is this then? A combination of all three methods? Sorry but I cant find this information, maybe I just didnt catch it?

"For displaying data per sample on the boxplots, we plotted the adjusted intensity values for each metabolite produced by the removeBatchEffects function in the limma R package using age, sex and cohort as covariate" I understand this but from which area ratio, coming from which analysis method? I mean some metabolites will pop up in ESI+ and ESI- others will pop up in both your ESI- methods and so on. I am sorry but this is still confusing, what have the authors really done? Where does the data in the figures come from?

My problem is: I cant really follow the correlations between how data was generated and which data was further used?

I appreciate the absolute quantification, nice to see and important. However, the authors say that they have thoroughly described the quantitation method. Frankly, none of my grad students could repeat this with the given details. I am not asking for a lengthy validation, but just a quick check of linearity would be important, did the authors construct calibration lines? Which of the 3 methods was used? The IS are added at 2 microM, yes, but which method was subsequently used? Was this also done on an Orbitrap, what ions were used for quantification...sorry but this is not thoroughly described. I mean this is not an analytical paper and I understand that, but if I cant repeat what is written here what is the point of it? Sorry, I dont want to be too harsh, these all are interesting findings, but I personally still find this confusing and not detailed in a reproducible way. However, reproducibility is key for the further use of your data, so methods should be described in a manner that they can be repeated by others.

Minor

p16 l354 capital I

p18 l385 of? of what?

Martin Giera

Response to Reviewer's Comments

Reviewer #1 Comments: *The authors have addressed my comments. One minor point is that the authors may want to refer in the discussion to previous work linking IFN, Kyn, AHR and IL-10. Rothhammer et al (Nat medicine 2016) showed that AHR expression is induced by IFN-I, and AHR mediates many of the physiological effects of kynurenine including IL-10 production and the control of peripheral and CNS inflammation (Rothhammer et al 2019, NRI).*

Response: Thanks for pointing us to the interesting paper and review. The original article in Nature Medicine is now mentioned in the Discussion of the revised manuscript.

Reviewer's comments: *The authors have improved their experimental section. However, and yes, this might be me, but I still dont think the described metabolomics experiments can be repeated with the given details. The authors indicate to have run their samples on 3 platforms and then? You combined the data? You ran data analysis separately and then ended up with the 91 metabolites significantly changed? You did rel. quant on all 3 data sets? So which is the data set I am looking at in your figures? Were different matrices run with different methods? How comparable is this then? A combination of all three methods? Sorry but I cant find this information, maybe I just didnt catch it? "For displaying data per sample on the boxplots, we plotted the adjusted intensity values for each metabolite produced by the removeBatchEffects function in the limma R package using age, sex and cohort as covariate" I understand this but from which area ratio, coming from which analysis method? I mean some metabolites will pop up in ESI+ and ESI- others will pop up in both your ESI- methods and so on. I am sorry but this is still confusing, what have the authors really done? Where does the data in the figures come from? My problem is: I cant really follow the correlations between how data was generated and which data was further used? I appreciate the absolute quantification, nice to see and important. However, the authors say that they have thoroughly described the quantitation method. Frankly, none of my grad students could repeat this with the given details. I am not asking for a lengthy validation, but just a quick check of linearity would be important, did the authors construct calibration lines? Which of the 3 methods was used? The IS are added at 2 microM, yes, but which method was subsequently used? Was this also done on an Orbitrap, what ions were used for quantification...sorry but this is not thoroughly described. I mean this is not an analytical paper and I understand that, but if I cant repeat what is written here what is the point of it? Sorry, I dont want to be too harsh, these all are interesting findings, but I personally still find this confusing and not detailed in a reproducible way. However, reproducibility is key for the further use of your data, so methods should be described in a manner that they can be repeated by others.*

Response: We appreciate the opportunity to provide an exhaustive description of the methods employed throughout this study, with enough details to repeat our work and analyses in other laboratories and ensure its full reproducibility.

First, our initial metabolomics analyses were performed via UHPLC-HRMS to identify statistically significant and consistent metabolic changes between individuals with and without trisomy 21. We performed these initial studies in two fully separate cohorts, which were analyzed both individually and in combination. We have revised the text thoroughly to reflect that each cohort was run and analyzed independently initially. Importantly, these individual analyses identified largely overlapping groups of metabolites as differentially abundant in people with Down syndrome, including numerous components of the KP. As described in the text, these individual analyses also identified different distributions of metabolite abundances between cohorts. Therefore, to identify consistent metabolic impacts and increase statistical power, we combined the cohorts by modeling age and sex as covariates and fitting each cohort as a batch effect. It was this combined analysis that yielded the 91 metabolites consistently measured across samples, of which 29 were differentially abundant. It was also this combined analysis that was used for the volcano plot in **Fig. 1a** as well as the box plots in **Fig. 1b-c**, **Supplementary Fig. 1d**, and **Supplementary Fig. 2a**. All of this information is now documented extensively in **Supplementary Data 1** (cohort information), and **Supplementary Data 2-3** (raw data for each cohort), and **Supplementary Data 4** (results of the individual and combined analyses).

Second, following a request by Reviewers, we employed a quantitative mass spectrometry approach that utilized stable-isotope labeled TRP and KYN with a modified extraction method designed to improve

chromatographic baseline separation and peak resolution resulting in improved quantification of TRP pathway metabolites^{1,2}. Combined with the use of stable isotope-labeled internal standards, these methods allowed us to perform absolute quantitation of tryptophan, kynurenine and other metabolites in this pathway. Calibration curves were generated (see for example the two curves in **Fig. R1** for Reviewers only for tryptophan and kynurenine over 4 orders of magnitude) across the concentration ranges relevant to the abundance of these metabolites in plasma or CSF. We applied this method to a larger cohort, referred to as Cohort 3 (for plasma analysis) and, in response to another request from Reviewers, to a fourth separate cohort for which CSF samples were available (Cohort 4). This absolute quantitation method is now fully documented in the revised manuscript in **Supplementary Data 1** (cohort information), **Supplementary Data 5-6** (absolute quantification of plasma) and **Supplementary Data 9-10** (absolute quantification of CSF). These methods are also described in detail in the revised **Methods** section.

Figure R1. Calibration curves for Tryptophan and Kynurenine from absolute quantification experiments.

These methods are also described in detail in the revised **Methods** section.

Third, following Reviewer's guidance, we now provide the information of which modality each metabolite was detected (see new **Supplementary Data 2-3**). As pointed out by the Reviewer, all of the metabolites identifiable in the isocratic-based method would also be quantifiable in the gradient-based methods - usually with high correlations between the measurements from both methods³. Therefore, in the end we ended up deriving quantitative measurements (both relative and absolute) from the 5-minute method⁴. We now describe these methods extensively in the manuscript. Furthermore, in the revised **Supplementary Data** we now report retention times and raw data from the 5-minute method runs and have uploaded these data to Metabolomics Workbench with Study IDs ST001240, ST001241, ST001242, and ST001243.

Fourth, all data and R code developed for the full set of statistical metabolomic analyses is available at: https://github.com/CostelloLab/Trisomy21_KYN_metabolomics. To ensure reproducibility, this code can be run to generate all figures for this manuscript. This information and weblink is provided in the **Methods** section.

We hope the revised text, the extended **Methods** section, the comprehensive **Supplementary Data**, the uploading of all data to **Metabolomics Workbench**, and the fully available **R code package** will address the Reviewers' concerns and make very clear to readers of the manuscript how the data was generated and analyzed.

Reviewer's comment: Minor: p16 l354 capital I; p18 l385 of? of what?

Response: We have revised the manuscript to correct these typos.

References.

- 1 Greene, L. I. *et al.* A Role for Tryptophan-2,3-dioxygenase in CD8 T-cell Suppression and Evidence of Tryptophan Catabolism in Breast Cancer Patient Plasma. *Molecular cancer research : MCR* **17**, 131-139, doi:10.1158/1541-7786.MCR-18-0362 (2019).
- 2 Rogers, T. J. *et al.* Reversal of Triple-Negative Breast Cancer EMT by miR-200c Decreases Tryptophan Catabolism and a Program of Immunosuppression. *Molecular cancer research : MCR* **17**, 30-41, doi:10.1158/1541-7786.MCR-18-0246 (2019).
- 3 Nemkov, T., Hansen, K. C. & D'Alessandro, A. A three-minute method for high-throughput quantitative metabolomics and quantitative tracing experiments of central carbon and nitrogen pathways. *Rapid Commun Mass Spectrom* **31**, 663-673, doi:10.1002/rcm.7834 (2017).

- 4 Nemkov, T., Reisz, J. A., Gehrke, S., Hansen, K. C. & D'Alessandro, A. High-Throughput Metabolomics: Isocratic and Gradient Mass Spectrometry-Based Methods. *Methods in molecular biology* **1978**, 13-26, doi:10.1007/978-1-4939-9236-2_2 (2019).

REVIEWERS' COMMENTS:

Reviewer #3 (Remarks to the Author):

The authors now provide an experimental description which is clear and can be followed. I hope their study will lead the way to clinically relevant recommendations and treatment for people with T21.

Martin Giera